# Investigation of Vehicle Stability with Consideration of Suspension Performance

**Vaidas Lukoševičius [1,\*], Rolandas Makaras [1], Arūnas Rutka [2], Robertas Keršys [1], Andrius Dargužis [1] and Ramūnas Skvireckas [1]**

1 Faculty of Mechanical Engineering and Design, Kaunas University of Technology, Studentų Str. 56, 51424 Kaunas, Lithuania; rolandas.makaras@ktu.lt (R.M.); robertas.kersys@ktu.lt (R.K.); andrius.darguzis@ktu.lt (A.D.); ramunas.skvireckas@ktu.lt (R.S.)
2 Lithuanian Road Administration, J. Basanavičiaus Str. 36, 03109 Vilnius, Lithuania; arunas.rutka@lakd.lt
\* Correspondence: vaidas.lukosevicius@ktu.lt

**Abstract:** The issue of movement stability remains highly relevant considering increasing vehicle speeds. The evaluation of vehicle stability parameters and the modeling of specific movement modes is a complex task, as no universal evaluation criteria have been established. The main task in modeling car stability is an integrated assessment of the vehicle's road interactions and identification of relationships. The main system affecting the vehicle's road interaction is the suspension of the vehicle. Vehicle suspension is required to provide constant wheel to road surface contact, thus creating the preconditions for stability of vehicle movement. At the same time, it must provide the maximum possible body insulation against the effect of unevennesses on the road surface. Combining the two marginal prerequisites is challenging, and the issue has not been definitively solved to this day. Inaccurate alignment of the suspension and damping characteristics of the vehicle suspension impairs the stability of the vehicle, and passengers feel discomfort due to increased vibrations of the vehicle body. As a result, the driving speed is artificially restricted, the durability of the vehicle body is reduced, and the transported cargo is affected. In the study, analytical computational and experimental research methods were used. Specialized vehicle-road interaction assessment programs were developed for theoretical investigation. The methodology developed for assessing vehicle movement stability may be used for the following purposes: design and improvement of vehicle suspension and other mechanisms that determine vehicle stability; analysis of road spans assigned with characteristic vehicle movement settings; road accident situation analysis; design of road structures and establishment of certain operational restrictions on the road structures. A vehicle suspension test bench that included original structure mechanisms that simulate the effect of the road surface was designed and manufactured to test the results of theoretical calculations describing the work of the vehicle suspension and to study various suspension parameters. Experimental investigations were carried out by examining the vibrations of vehicle suspension elements caused by unevenness on the road surface.

**Keywords:** quarter-car model; road unevenness; suspension performance; suspension test bench; vehicle stability





## 1. Introduction

Vehicle stability depends on a number of factors, and a general investigation of stability would not accurately reflect the effect of individual systems. Vehicle-road interaction and the factors that affect this relationship have the greatest impact on vehicle stability. The key elements of the interaction are the following: the structure of the vehicle suspension and the changes in wheel position influenced by it, the nature of the road surface, the structure, and the principles of operation of the elastic and damping elements. Most applied vehicle stability research is dedicated to the study and improvement of individual elements of vehicle-road interaction.

The main assembly affecting vehicle-road interaction is the vehicle suspension. Designing suspension of a completely new type would be a highly difficult task, so the maximum scientific capacity and research base must be dedicated to improving conventional structures. Fewer cars feature classic passive suspensions. Various control systems with active elements that adjust the suspension properties as necessary are becoming increasingly popular. Anti-lock braking, anti-skid systems, and four-wheel steering systems have become popular in recent decades. All of the above factors determine the stability of vehicle movement.

The unevenness and roughness of the road surface are the key environmental factors affecting a moving vehicle. Therefore, it is crucial to accurately assess the dynamic response of the vehicle that is subject to the most realistic road model possible. However, the development of the latter presents a great challenge. Road models are usually expressed as a function of spectral density that accounts for the rises and falls of a road profile. In road unevenness models, the idealized road profile is considered to be a random process, and random deviations are not analyzed.

The work of M.W. Sayers, T.D. Gillespie, and S.M. Karamihas at the University of Michigan Transportation Research Institute should be emphasized in an analysis of road surface assessment. This research work [1–3] is among the key works that have contributed to the establishment of globally recognized principles of road surface measurement and data processing criteria. In this work, the researchers investigated methods and equipment for measuring various parameters of the road surface and provided comprehensive criteria for the determination of IRI (International Roughness Index). IRI measurement is based on an assessment of the effect of the road surface on the parameters established for the quarter-car model. Other studies [4–9] can also be considered, since they evaluate the effect of the road surface on the vehicle.

The tire is another important element that influences vehicle stability. Pneumatic tires are essentially complex force generators. This means that only complex algebraic expressions enable the exact modeling of the tires. In their paper [10], M. EI-Gindy and H. Lewis present an extensive investigation of the contact area of the tire-road surface. Several other researchers have also evaluated the effect of the tire [11–18]. H.B. Pacejka from TU Delft performed important research in the field of vehicle stability, performance variables, and tyre modelling. He investigated the nonlinear motion of the car during sudden manoeuvres, modelling the behaviour of the car during sudden acceleration or braking. This research works dealt with various wheel-slip situations and provided model simulations of longitudinal and lateral slip [19].

Research articles have primarily investigated the design and improvement of active suspension. The use of an active suspension system enables the solution of many complex suspension design issues. Designers of active systems have provided significant contribution to the development of suspensions with different levels of activity: semi-active, adaptive, active damping, and other types. In individual cases, these suspensions can be compared to fully active ones, in particular when considering a simpler design for reliability. The main research objects considered when dealing with suspension work are single or two-mass quarter-car models. The synthesis criteria and optimization methods of control theory are applied to the analysis of these models. Accurate assessment of an active car suspension is possible only with the use of more complex two or three-dimensional car models. The accuracy of these models is further enhanced by the lateral tilt and turns of the entire vehicle. The most common research works focus on the separate analysis of dependencies: the delay of the excitation effect between the front and rear wheels, differences between parallel rolling wheels on one axle and deformability of the car frame, among others.

An article published by J. Wattan, K.M. Holford, P. Surwattanawan [20] is insightful as it deals with the issues of active suspension modeling and employs a real-life quarter-car model described in the article. Other articles deal with active suspension control problems [21–24] or analyze the performance of specific suspensions [25–27].

Some research investigates the active suspension anticipation function. Two main control schemes are used most often. The one located in front of the data vehicle uses the obtained data in the control system that is designed to eliminate the excitatory effect of the road surface. The performance of another system is based on an assessment of the road surface by examining the dynamics of the front axle. Studies have shown that the control system is highly dependent on the quality of the road surface. Theoretical calculations provide for improvements of the active suspension and account for the applicability of the anticipation function.

Research by D. Horvat [28,29] analyzed the methods of road surface assessment and model the vehicle movement. The research was performed using quarter-car and spatial models of a vehicle with a different number of degrees of freedom. The main criterion for assessing the impact of the road surface is the acceleration of the sprung mass of the vehicle. The properties of active suspensions may be provided to semiactive suspension systems by using components similar to passive suspensions. The main idea is to use an additional elastic element mounted parallel to the damper, the properties of which can be adjusted to high-frequency performance. A semi-active damper is an externally controlled force generator. Its capacities are limited, as the stored energy must be dissipated.

A separate research field is dedicated to the investigation of the design of various suspension elements and possibilities for their improvement. Considerable research efforts have been made to improve suspension elements and their properties [30–35]. Certain articles deal with the vibrations of various vehicle elements and the methods of their measurement [36–38]. Very few research works provide a generalized analysis of different types of car suspension design. The German researcher J. Reimpell is one of the most prominent world-class experts in this field. His research papers have been published all over the world and translated into different national languages [39–43]. His work provides a comprehensive investigation of vehicle suspension structures and discussion of the principles and methods of calculation of the characteristic parameters of suspension.

The designs of suspension systems and their main characteristic parameters have been investigated by T. D. Gillespie [44], W. Matschinsky [45] and D. Bastow [46]. In their studies, they analyze different types of suspension and discuss key parameters that affect the movement and stability of a vehicle.

Reviews have confirmed that efforts are being made to make use of the rigid and damping elements of suspensions with variable characteristics in an attempt to combine conflicting requirements such as driving comfort and directional stability of the vehicle. However, there are no comprehensive data on the performance of these systems for poorer quality roads.

This article reviews the road surface and vehicle tire interaction models that include real road surface properties. Classification of vehicle suspensions according to summarized kinematic suspension properties that provide the prerequisites for the development of universal models that account for the performance of suspensions with diverse kinematic properties are proposed. Models that account for the effect of the suspension on the vehicle movement stability were developed. A vehicle suspension test bench that includes the mechanisms of the original design to simulate the effect of the road surface was designed and manufactured to verify the results of the theoretical calculations describing the performance of the vehicle suspension and to study various suspension parameters.

Based on the topics above, the main contributions of this paper are: (1) a description of the road surface is revised and provides specifications of the interaction of road surface-tire contact areas and the equalizing properties of the tire; (2) a dynamic quarter-car model is developed and accounts for the effect of suspension, and the effect of the individual suspension elements on the characteristics that determine movement stability is determined, with the friction force originating from the transverse displacement of the wheel in the damper frame and the variation thereof being assessed; (3) the applicability of a vertical dynamics model is verified by experiments, with experimental investigations being conducted and the vibrations of the vehicle suspension elements caused by the road surface

unevennesses explored using the developed vehicle suspension test bench that simulates driving conditions and replicating the real road surface effect; (4) the effect of geometric position changes of the wheel on the parameter of directional stability of the vehicle, i.e., on the slip angle, is determined using kinematic models of the vehicle suspension. The developed software application is used to calculate the effect of the above factors on the dependence of the vehicle limit turning radius on the speed.

## 2. Vehicle Stability Modeling Methods

Trend analysis of the traffic and vehicle design patterns suggests that handling, i.e., the ability to move in the direction intended by the driver, is one of the key performance properties of vehicles that influence their safety. The driver sets the intended direction using the steering wheel, and the handling tasks are targeted at the steering mechanism. One of the first tasks addressed was the interrelation between the front wheel angles. The Ackermann-Jeantaud scheme (Figure 1) was used for this purpose.

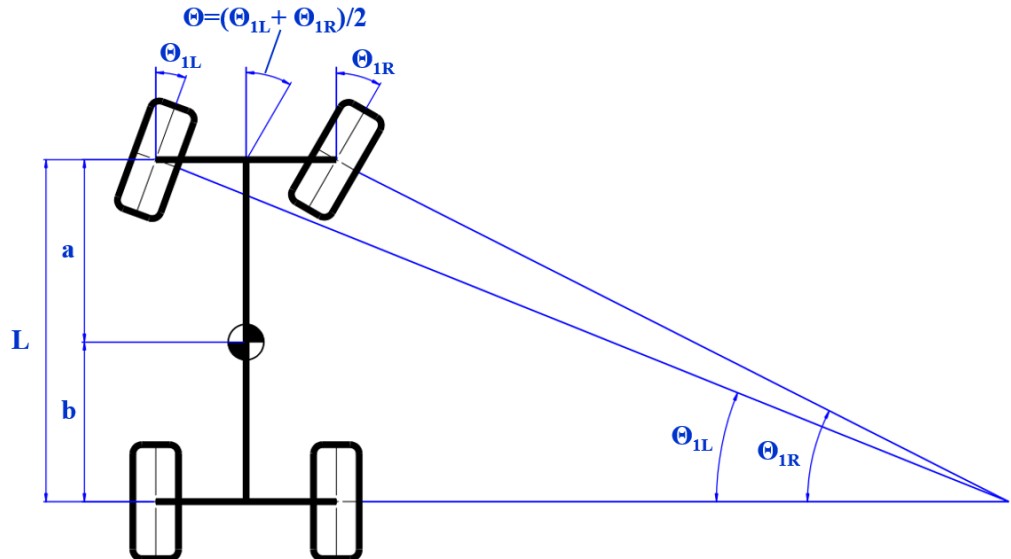

**Figure 1.** Ackermann—Jeantaud schema [47].

Following the introduction of pneumatic tires into modeling, the dependencies of the tire-road interrelation had to be analyzed in greater detail. The area of contact between the tire and the road is subject to deformation under the action of a lateral force. As a result, the contact patch axis is no longer parallel to the wheel symmetry plane, but forms angle $\delta$, the former being referred to as the tire slip. In most tasks, the slip angle is determined as the angle between the velocities longitudinally and transversely to the wheel symmetry plane. The lateral force $F_y$ that the wheel has the capacity to withstand depends on the radial load and slip angle. Tire slip angle (Figure 2) causes a considerable distortion of the cornering scheme [48]. There are several theories that can be used to determine the angle of tire-slip. The theory of M. V. Keldysh [48] is most convenient. Under M. V. Keldysh's theory, lateral tire deformation $\xi$ and trajectory curvature $\rho$ are described by the following Equations [47,48]:

$$F_y = C_y\xi, \frac{1}{\rho} = \alpha\xi - \beta\varphi_t \tag{1}$$

where $C_y$ is the lateral stiffness of the tire, $\rho$ is the trajectory curvature radius, $\alpha$ and $\beta$ are the constant coefficients associated with the tire structure, and $\varphi_t$ is the angle between the wheel symmetry plane and wheel trajectory tangent (Figure 2).

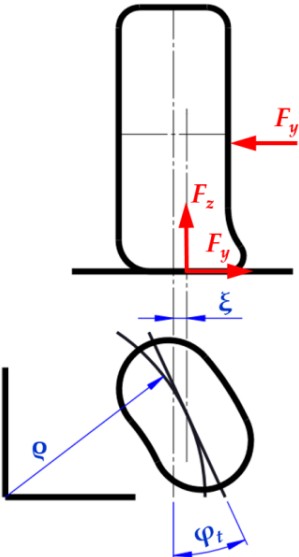

**Figure 2.** Tire deformation scheme in the presence of slip [48].

If $F_y \neq$ const., then the wheel center movement velocity in the transverse direction equals is given by:

$$v_y = v\varphi_t + \dot{\zeta} \tag{2}$$

where $v_y/v = \delta$, then $\delta = \varphi_t + \dot{\zeta}/v$, from which it follows:

$$\varphi_t = \delta - \dot{\zeta}/v \tag{3}$$

This would lead to the condition that only the tire slip angle would affect the vehicle movement direction, the prerequisite being $\dot{\zeta} = 0$, i.e., $F_y$ = const.

For low $\varphi_t$, the wheel trajectory is straight, and the contact patch area is positioned at angle $\varphi_t = \delta$ relative to the symmetry plane of the wheel. With the available inertia forces, the longitudinal and transverse movement of the vehicle can be described. Considering that $\theta$, $\delta_1$ and $\delta_2$ are low, and their cosines are equal to each other, while sines are equal to the angle values (Figure 3), and forming the expression $F_{y1} = k_{y1}\delta_1$ and $F_{y2} = k_{y2}\delta_2$, the transverse force equilibrium is described by Equation (4) [48]:

$$F_{iy} = m_a(v\omega_a + \dot{v}_y) = k_{y1}\delta_1 + k_{y2}\delta_2 - F_{x1}\Theta \tag{4}$$

where $m_a$ is the vehicle center of mass; $\omega_a$ is the vehicle angular velocity when cornering; $k_{y1}$ and $k_{y2}$ are the slip angle drag coefficients; $F_{x1}$ and $F_{y1}$, $F_{x2}$ and $F_{y2}$ are drag forces for the front and rear wheels respectively and $\Theta$ is the wheel turning angle.

Following expression of $\delta_1$ and $\delta_2$ by using $v_y$ and $\omega_a$:

$$\delta_1 = \Theta - (a\omega_a + v_y)/v, \ \delta_2 = (b\omega_a - v_y)/v \tag{5}$$

Following input of the $\delta_1$ and $\delta_2$ values into Equation (5), the following is obtained:

$$\dot{v}_y + v_y(k_{y1} + k_{y2})/(m_a v) + \omega_a[v + (k_{y1}a - k_{y2}b)/(m_a v)] - k_{y1}\Theta/m_a = 0 \tag{6}$$

By applying the equation of the moments at the center of mass $I_z\dot{\omega}_z = k_{y1}a - k_{y2}b$ and identifying variables $v$ and $\omega$, the following is obtained:

$$\ddot{v}_y + m\dot{v}_y + pv_y = q_{y1}\dot{\Theta} + q_{y2}\Theta, \ \ddot{\omega}_a + m\dot{\omega}_a + p\omega = q_{\omega1}\dot{\Theta} + q_{\omega2}\Theta \tag{7}$$

where $m$, $p$, $q_{y1}$, $q_{y2}$, $q_{\omega1}$, $q_{\omega2}$ are the relative drag coefficients of the front and rear axles.

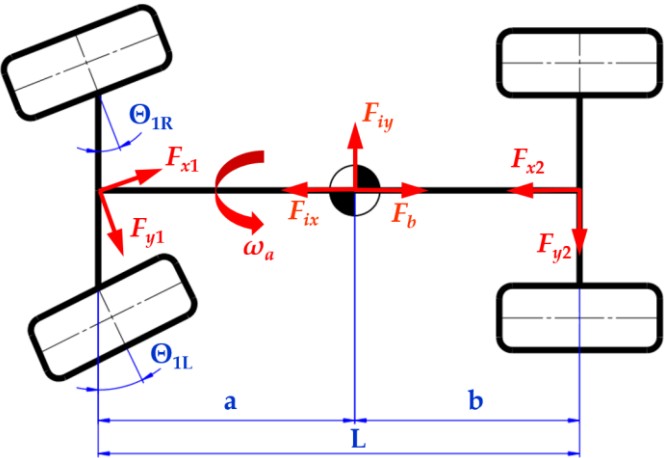

**Figure 3.** Forces acting at vehicle cornering [48].

The dependence of the longitudinal and lateral parameters of the vehicle on the steerable wheel turn angle can be determined by solving Equations (6) and (7), and their dependence on the steering wheel turn angle can be determined if the steering ratio value is available. Equations (6) and (7) can be expressed in a different form [49]:

$$I_z \dddot{\psi} + D\ddot{\psi} + C\dot{\psi} = \dot{M}_z + HM_z + k_{y1}a\dot{\delta}_1 + E\delta_1 + k_{y2}b\dot{\delta}_2 + E\delta \tag{8}$$

where $I_z$ is the moment of inertia about the vertical axis; $M_z$ is the external moment about the vertical axis and $\psi$ is the angle of the turn about the moment center. Coefficients $D$, $H$, $C$, and $E$ are calculated as follows:

$$D = \frac{I_z(k_{y1}+k_{y2})}{m_a v_c} + \frac{k_{y1}a^2 + k_{y2}b^2}{v_c}, \ H = \frac{k_{y1}+k_{y2}}{m_a v_c}, \ C = \frac{L^2 k_{y1} k_{y2}}{m_a v_c} - k_{y1}a + k_{y2}b,$$

$$E = \frac{L k_{y1} k_{y2}}{m_a v_c} \tag{9}$$

where $v_c$ is the velocity of the center of mass (its direction does not correspond to the longitudinal axis of the vehicle).

The following is obtained:

$$I_z \dddot{\psi} + D\ddot{\psi} + C\dot{\psi} = \Phi(M_z, \delta_1, \delta_2) \tag{10}$$

where $\Phi$ is the function of effects.

The stability of the solutions of Equation (10) depends on the coefficients of the equation and the effect function $\Phi$. If the function of effects were not considered, the vehicle movement would be linear, and its stability would depend on coefficients $D$ and $C$.

Although designers may solve the problem and implement respective measures, the spatial position of the wheel changes at excitation, for example, due to road unevenness, and this may lead to a change in vehicle direction. More comprehensive studies have shown that slip angle characteristics are not enough, as the direction vector of the wheel is affected by the spatial position of the wheel, i.e., camber, alignment, etc. Direct application of dynamic equations is impossible due to the specifics of the tasks. Furthermore, the characteristics of individual elements are clearly nonlinear. Semiempirical formulas are used to describe the effect of the forces acting on the wheel and the spatial position of the wheel on the direction vector. Modeling requires dependences of the interrelation between the slip angle and lateral force, between the lateral force and relative wheel slip, between the slip angle and stabilizing moment. The analysis carried out has suggested that the currently available methodologies applicable to the factors affecting directional stability employ simplified models that individually address steering mechanism kinematics, suspension kinematics, suspension element stiffness, and forces acting in vehicle cornering. This kind

of analysis does not enable the identification of key factors. Therefore, to perform an integrated analysis, models consisting of multiple elements must be used. However, the characteristics of the elements can only be identified if detailed information is available on the specific vehicle design. This creates difficulties when performing an integrated analysis and drawing general conclusions in an investigation of directional stability as a criterion for the assessment of the technical condition of a vehicle and the condition of the road surface.

## 3. Investigation of the Factors Affecting the Vehicle Stability

### 3.1. Road Description

Records of the actual Lithuanian road microprofile were used in modeling the vehicle stability tasks. Records were generated by road surface measurements performed using the DYNATEST 5051 RSP (Dynatest A/S, Ballerup, Denmark) profilometer mounted on the VW Transporter [18,49]. Whereas road profile records are characterized by exception and span-specific properties, the possibilities for formalization of the road microprofile description were analyzed by identifying the typical features of road surfaces of different quality. During formalization of the road description, the discrete nature and large interval of the road microprofile records, was considered. For example, the information on the micro-irregularity height within a single established line is recorded every 0.147 m under the IRI calculation methodology and every 1 m when spatial road microprofile is recorded. Road microprofile records contain road surface data on the two longitudinal lines that correspond to the vehicle wheel rolling surface, and the longitudinal distance between the data is 0.147 m. Analysis of available records confirmed that essential differences can be assessed using generalized data of the road span microprofile for an even asphalt pavement, uneven asphalt pavement (in service), and gravel road surfaces. The microprofile records of the road spans were analyzed using classic methodology by calculating the dispersion of unevennesses $\sigma_q^2$ and correlation function $R_q$:

$$\sigma_q{}^2 = \lim_{L_q \to \infty} \frac{1}{L_q} \int_0^{L_q} q_0{}^2(x)dx, \ R_q(x_s) = \frac{1}{L_q \sigma_q{}^2} \int_0^{L_q} q_0{}^2(x)q_0{}^2(x - x_s)dx \tag{11}$$

where $L_q$ is the span length; $q_0$ is the microprofile height with the $y$–coordinate of the midline equal to zero interval $x_s$.

Road surface roughness data were used for an accurate assessment of the effect of the road surface and investigation of the equalization function of the tire. The data reflect the roughness of the road surface on a single line every 2.5 mm. Data from three different road surfaces were used for the investigation: relatively even (I), medium (II) and very rough (III). These microprofile records are fairly accurate, and the effect of roughness on the road surface was considered in addition. The road surface roughness span data was distributed in the microprofile according to the principle of a random number generator. These measures enabled the authors to fairly accurately reflect the key characteristics of the road surface. The data on the microprofile and road surface roughness characteristics of the investigated road spans are presented in Table 1. The characteristic wavelengths were revised by harmonic analysis involving identification of the lengths of maximum amplitude waves $R_i = \sqrt{a_i^2 + b_i^2}$ for Equation (12):

$$q(x) = \frac{x_0}{2} + \sum_{i=1}^{\infty} a_i \cos(i\omega x) + \sum_{i=1}^{\infty} b_i \cos(i\omega x) \tag{12}$$

**Table 1.** Microprofile and roughness characteristics of the investigated road surfaces.

| Road Surface Type | Dispersion $\sigma_q$, mm | Characteristic Wavelength, m |
|---|---|---|
| Even asphalt pavement | 3.492 | 50–52 |
| Uneven asphalt pavement | 12.437 | 75–77 |
| Gravel road | 13.667 | 45–47 |
| Roughness I (relatively even) | 0.276 | 0.04–0.1 |
| Roughness II (medium) | 0.695 | – |
| Roughness II (very rough) | 1.667 | 0.06–0.14 |

The waviness of the road surface and its amplitudes were determined following harmonic analysis of typical road microprofile fragments. The maximum amplitude waves were longer than 12 m in the cases analyzed. Only a few longer waves were found. The number of shorter waves was higher, but the amplitude of the shorter waves was very small at 0.3 to 1 mm depending on the microprofile evenness. It should be emphasized that short wavelength unevenness amplitudes were clearly dependent on the road surface type. For modeling using a flat model, the effect of the road surface was assessed by analyzing the microprofile points on a single longitudinal line.

The correlation functions of the road surface roughness are presented in Figure 4. It was noticed that waviness with waves shorter than those of the microprofile was characteristic of rough road surfaces.

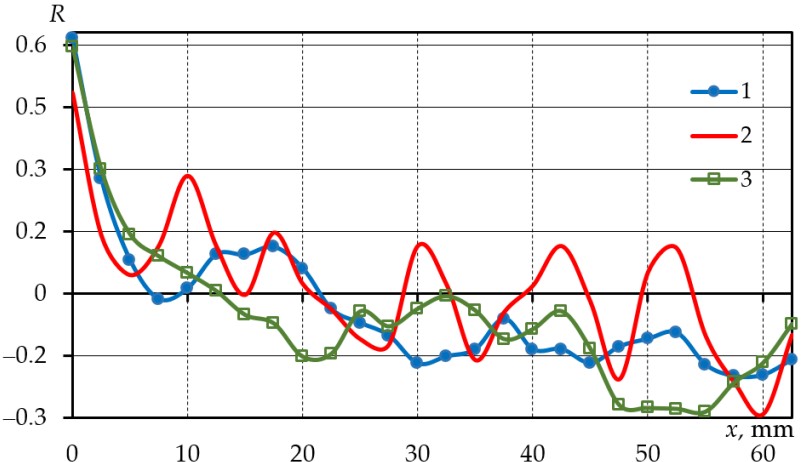

**Figure 4.** Correlation functions of road surfaces of different roughness: 1 = even, 2 = medium, 3 = very rough.

The spatial record of the road microprofile was used to determine the characteristics of the transverse road and to accurately model vehicle movement. Data from the 3 m wide road lane microprofile were recorded at nine points in the transverse, longitudinal direction, with a 1 m interval between the data. The description of the road span microprofiles was facilitated by the fact that the microprofile records generated by the DYNATEST 5051 RPS profilometer were densified in the spatial (3D) mode in the ruts to reproduce the microprofile specifics of the tire-road contact area more accurately.

The spatial microprofile record analysis showed waviness (Figure 5) that was characterized by perpendicularity, or a slight tilt relative to the axis, and relatively low transverse waves. The analysis of the spatial road surface roughness fragment (Figure 6) suggested that the roughness was a random set of unevennesses, having a certain prevailing shape of unevennesses.

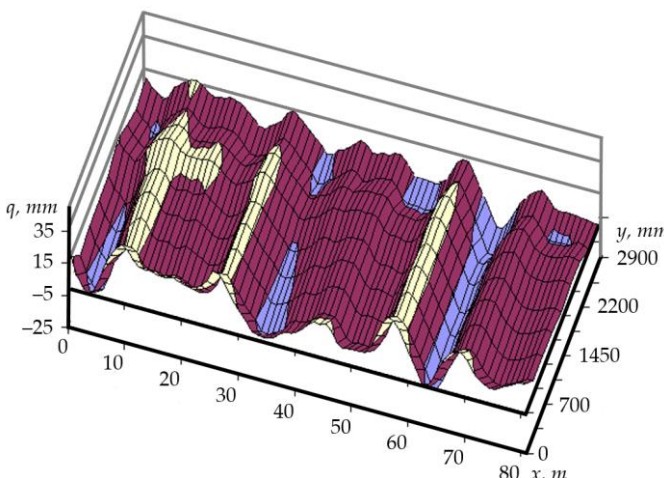

**Figure 5.** Spatial fragment of a road surface microprofile span.

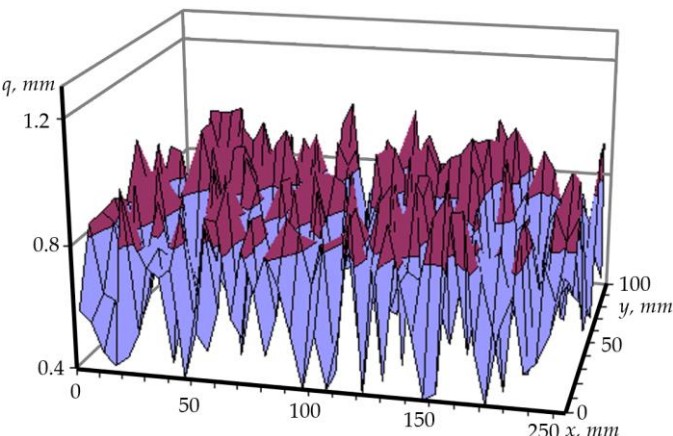

**Figure 6.** Fragment of a spatial road surface roughness.

In the case of spatial modeling, it is necessary to assess the nature of the road surface in the transverse plane relative to the vehicle movement. The spatial road microprofile record was used as the baseline in the calculation application and accounted for the nature of the variation of the microprofile in the longitudinal and transverse directions. These data provided accurate information on the specifics of the road surface in the transverse directions (ruts, camber, banks). A microprofile of a smaller interval was reproduced according to more detailed records. Road surface roughness was also assessed during modeling. The calculation application includes the option to assess random damages to the road surface, e.g., cracks present in the road surface.

### 3.2. Investigation of the Equalizing Function of the Tire

During modeling of vehicle wheel rolling on a road span with determined characteristics, assessment of road-tire interaction is important even where the road microprofile record is available. Two aspects were analyzed during the study. The first was related to the fact that a rolling wheel would not copy the unevenness of the road surface, but would rather be characterized by an equalizing effect. This had to be taken into account in the case of description of movement on a very uneven road. The second aspect was related to tire deformation. Theoretically, assuming that the tire deformability is linear, it needs to be described in the tire description as a nonlinear element. This is due to the constant variation of the tire-road contact area and nonlinear dependence between the contact dimensions (e.g., contact length) and vertical deformation during deformation of the tire.

During the study, the effect of the tire on vehicle-road interaction was investigated to select the simplest possible tire model that would be characterized by sufficient precision. The study was limited to linear and flat tire-road contact assessment irrespective of the tire structure (quasi-static model). Flexible narrow ring (2D) and flexible band (3D) models were chosen for a more detailed investigation of the equalizing function of the tire. The former (Figure 7a) assesses the linear tire-road contact and is used where a simplified microprofile record is available, i.e., the road microprofile on a single line. The latter (Figure 7b) assesses the spatial contact between the tires and the road, but requires a road microprofile record that reflects the condition of the entire traffic lane. During the application of the models referred to above, the contact length of the tire was determined according to the vertical deformation of the tire. Two marginal cases were analyzed: (i) tire deformation in the circular direction, which was absolutely absent with contact length equal to arc length (this assumption was inaccurate, as experimental data confirmed that the tire was subject to compression in the contact area), and (ii) compression of the tire tread up to the chord length. The calculations suggested that the methodology of calculation of the contact length did not have a significant effect on the results, where the rolling of contemporary tires on roads in good condition was analyzed. In the calculations, a deterministic research approach was employed. For this purpose, real road microprofile records were formed on a single line for the 2D tire model, and spatial road microprofile records were formed for the 3D tire model. The roughness of the road surface was assessed by distributing the road surface data on the road microprofile. The road surface roughness data (control points located every 2.5 mm) were selected from the baseline data file according to the random number generator principle. This provided the calculated road microprofile data $q(x)$ or $q(x,y)$.

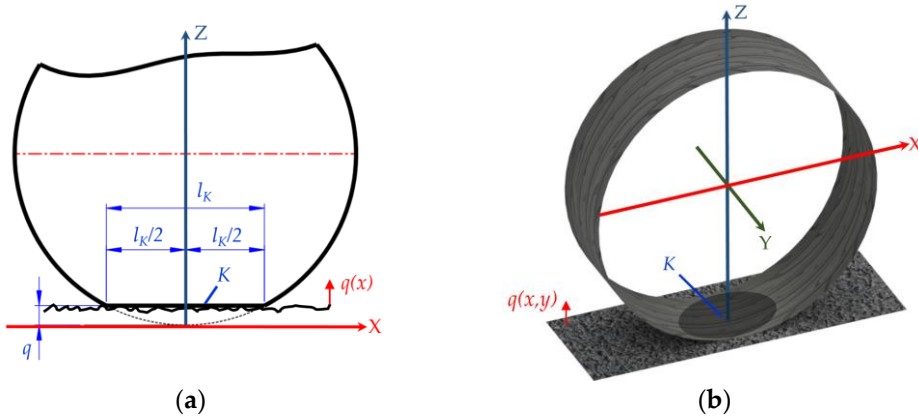

(**a**)　　　　　　　　　　　　　　　　　　　(**b**)

**Figure 7.** Tire models: (**a**) flexible ring; (**b**) flexible band.

The quarter-car model corresponding to the parameters of a city car was used in the calculations [18]. By using this model, the limits of variation in tire contact length were determined for roads of different quality (Figure 8). For roads with asphalt pavement, the length of the tire-road contact varied by about 16%, and about 15% of the length of the road contact was close to the length under static load. For an uneven road (Figure 8), the contact length curve was distributed over a much wider range. During the investigations, the cases where the tire became detached from the road, and the maximum allowable deformation of the tire was reached, were recorded. The analysis found that the role of the tire was more pronounced when modeling driving on very uneven roads. If necessary, the radii of the curvature of the road surface may be estimated at individual points of the profile. This kind of data processing significantly reduces the size of the initial data files and the duration of further calculations. Investigations showed that it is unreasonable to use the 3D model where there is no need to have data on vehicle wheel tilt or contact curvature in the transverse plane relative to wheel rolling.

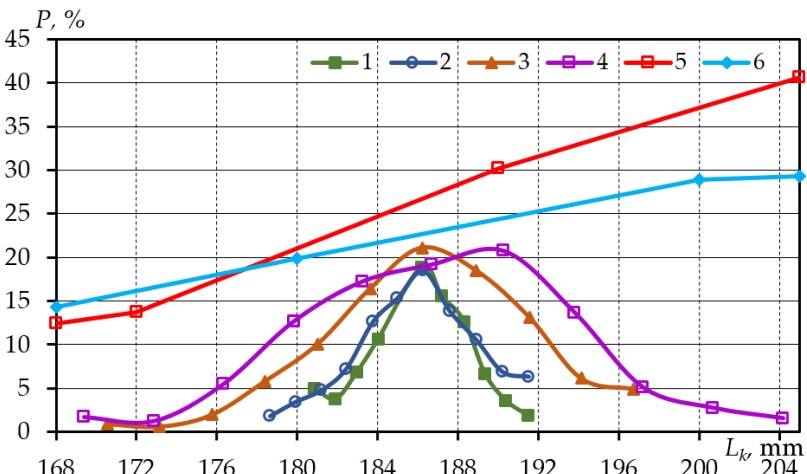

**Figure 8.** Tire contact length distribution for different road quality and tire models: 1—smooth asphalt pavement 2D; 2—smooth asphalt pavement 3D; 3—low quality asphalt pavement 2D; 4—low quality asphalt pavement 3D; 5—gravel road 2D; 6—gravel road 3D.

The vertical wheel displacement required for the evaluation of suspension performance is determined by the tire model with point contact. The results of the analysis of the microprofile records for roads of different quality suggested that the IRI measurement methodology proposed by M. Sayers [1,2] is solid when choosing the profile measurement criteria and the unevenness assessment interval.

### 3.3. Dynamic Quarter-Car Model

The quarter-car model was used for the tire-road interaction analysis. The model corresponded to the operating conditions of a single vehicle wheel and enabled researchers to simplify the analysis of vehicle dynamics. The quarter-car model is usually used to identify key vehicle suspension parameters and investigate tire performance.

Although the quarter-car model only covered the vertical elastic and damping suspension elements, the results enabled analysis of the directional stability of the vehicle at vertical excitation. The data obtained were used as the baseline data to identify the exact spatial position of the wheel (Section 5).

A revised quarter-car model (Figure 9) was used for the analysis of vehicle movement on road surfaces of different quality. In contrast to the two-mass model, the revised model also accounted for the tensile properties of the tire tread and the friction in the suspension elements. This model could provide more accurate structural parameters of the vehicle analyzed.

Model movement Equations for the vertical direction:

$$\begin{aligned}
&m_1\ddot{z}_1 + k_2(\dot{z}_1 - \dot{z}_2) + k_1(\dot{z}_1 - \dot{q}) + c_2(z_1 - z_2) + c_1(z_1 - q) = 0, \\
&m_2\ddot{z}_2 + k_3(\dot{z}_2 - \dot{z}_3) + k_2(\dot{z}_2 - \dot{z}_1) + c_3(z_2 - z_3) + c_2(z_2 - z_1) + F_\mu \mathrm{sgn}(\dot{z}_2 - \dot{z}_3) = 0, \\
&m_3\ddot{z}_3 + k_3(\dot{z}_3 - \dot{z}_2) + c_3(z_3 - z_2) + F_\mu \mathrm{sgn}(\dot{z}_3 - \dot{z}_2) = 0
\end{aligned} \tag{13}$$

where $m_1$, $m_2$, $m_3$ are the tread part of the tire, unsprung, and sprung mass, respectively; $z_1$, $z_2$, $z_3$ are the tire, suspension and body displacements, respectively; $k_1$, $k_2$, $k_3$ are the tread, tire, and suspension damping factors, respectively; $c_1$, $c_2$, $c_3$ are the stiffness of the tread, tire, and suspension, respectively; $F_\mu$ is the friction force at the shock absorber rod and q is the height of road profile unevenness.

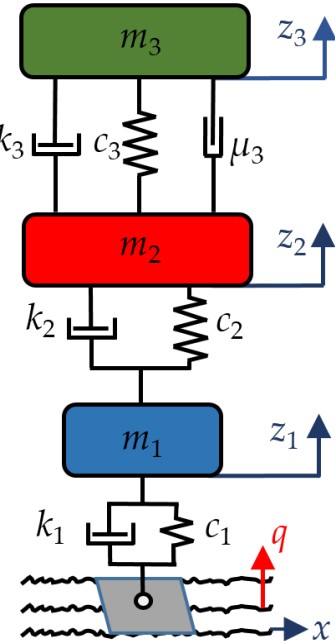

**Figure 9.** Quarter-car model.

When refining the quarter-car model, it is necessary to consider that the elastic elements and dampers of the suspension have distinctly nonlinear characteristics. In the investigated case, the characteristic of the front suspension spring of the modeled VW Golf was linear, as refined by the experiment. However, it must be taken into account that elastic travel stops block with nonlinear characteristics and start to act in extreme positions with respect to the car's suspension recoil and compression travel. Therefore, the overall characteristic of all the elastic elements of the suspension is close to typical (Figure 10a). In the model, the suspension characteristics are described by three lines. For the avoidance of uncertainty, additional refinements were used in the inflection zones of the characteristics. The damping characteristics of the damper were described in a similar way (Figure 10b).

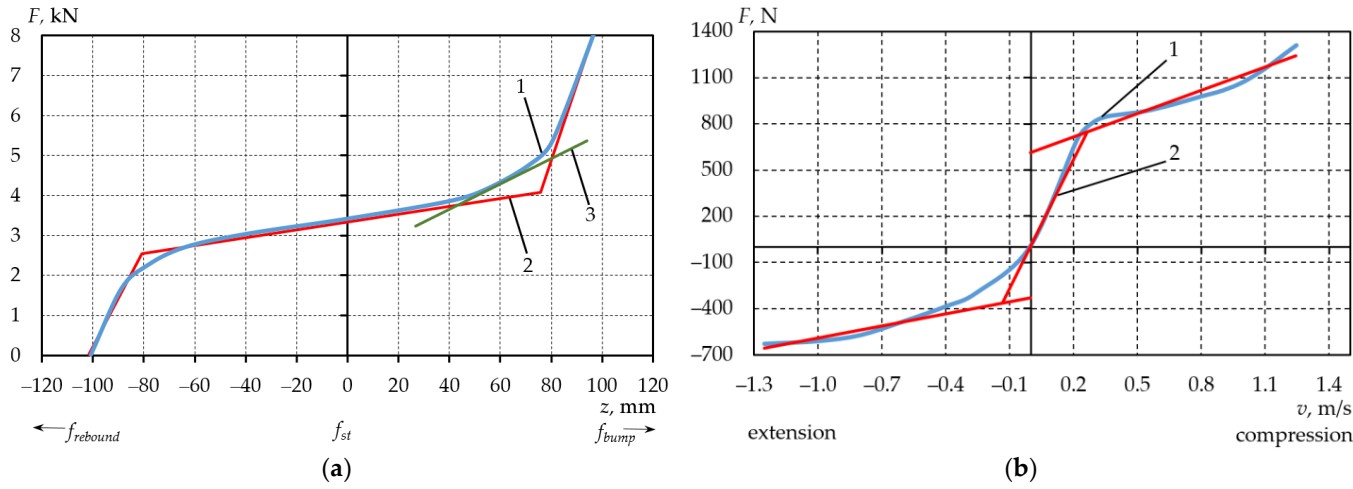

**Figure 10.** Modeling of suspension characteristics: (**a**)—elastic characteristic; (**b**)—damping characteristic of the damper; 1—typical characteristic; 2—simulated characteristic; 3—inflection zone refinement.

The frequency characteristic of the vehicle was determined by numerical modeling using calculation applications, with the road profile being sinusoidal in shape. The developed computer application enabled the analysis of vehicle movement with the set

parameters related to the road surface at the respective frequencies. The movement of the model corresponding to the city car parameters was investigated to determine its frequency characteristics (Figure 11). When compared to the theoretical frequency characteristic, low-frequency resonance oscillations of higher amplitude sprung masses of the body were observed. The frequency of resonance oscillations of the unsprung masses was lower than that determined by the theoretical calculations.

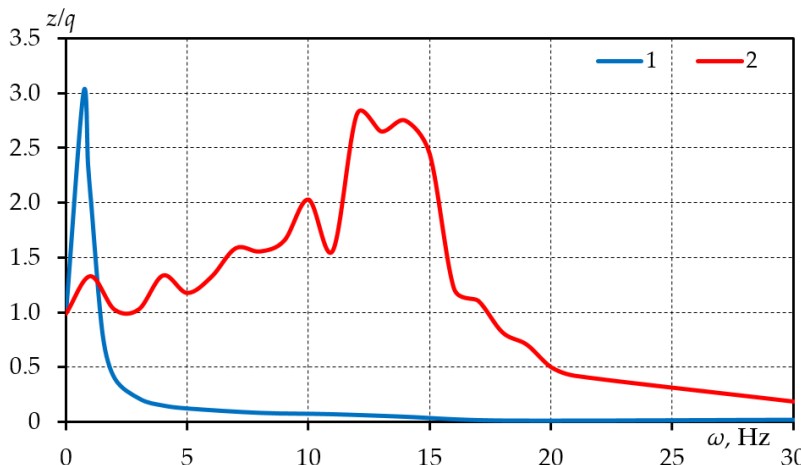

**Figure 11.** The frequency characteristic of the vehicle was determined with a numerical experiment: 1—sprung mass; 2—unsprung mass.

Calculations were performed to evaluate the uneven excitation effect of the road surface unevenness at various amplitudes on the frequency characteristic of the vehicle. During the evaluation of the effect of excitation amplitude, high-frequency resonance oscillation amplitudes of the unsprung mass were observed to decrease as the unevenness amplitude increased (Figure 12). However, the amplitudes of the low-frequency resonance oscillations increased in the latter case.

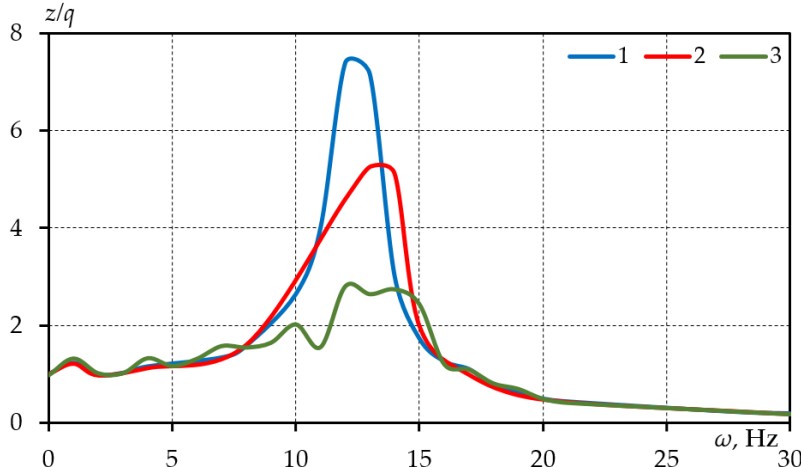

**Figure 12.** Frequency characteristic of the unsprung mass of the vehicle for the following road surface excitation amplitudes: 1—10 mm; 2—20 mm; 3—40 mm.

The effect of suspension characteristics on vehicle vertical displacements was determined by investigating the movement of the microprofile of the quarter-car model for the road surface microprofile of various parameters. A computer application was developed for the investigations. The application analyzed the vertical displacements of a three-mass quarter-car model and the effect of the nonlinear characteristics of the suspension.

Driving on one line was simulated without taking into account the transverse displacement of the wheel. The nonlinear characteristics of the suspension were found to mainly influence the movements of the sprung mass of the vehicle body (Figure 13). Simulation of driving on different road surfaces showed that the evaluation of the stiffness of the tire tread did not have a significant effect in most cases at low-frequency vibrations.

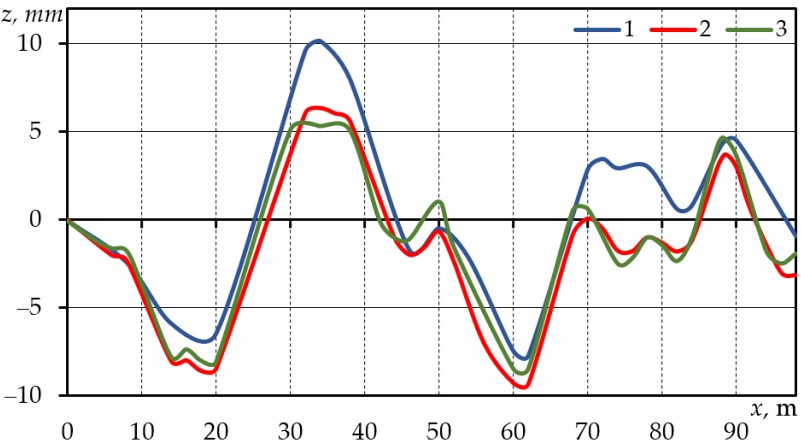

**Figure 13.** Vertical displacements of the sprung mass of the vehicle for driving on a smooth asphalt pavement road surface taking into account the effect of nonlinear elements: 1—linear elements; 2—nonlinear suspension characteristics; 3—effect of friction on the damper frame.

## 4. Experimental Investigations of Suspension Properties

### 4.1. Suspension Test Bench and Road Effect Simulation Equipment

A real quarter-car model was produced for the analysis of the theoretical calculations describing the performance of the vehicle suspension and for the investigation of the suspension parameters. This model enables investigation of vehicle suspension performance under laboratory conditions, different modes of movement, varying load conditions, and the excitation effect of different road surfaces.

The option of producing the car suspension test bench was chosen for the following main reasons:

- The car suspension test bench is not affected by random environmental conditions acting on the moving vehicle and the oscillations caused by the engine. This enables multiple experiments to be conducted under identical conditions. Therefore, this method provides more benefits compared to in-situ investigations of a real vehicle, where it is difficult to provide identical environmental conditions. Laboratory conditions allow for the use of stationary test equipment, thus simplifying the investigation process and reducing the duration of experiments.
- The test bench consists of the quarter car, i.e., the suspension of only one wheel is investigated. This makes it possible to investigate the performance of an individual wheel and its suspension elements without considering the remaining wheels or the movement of the entire vehicle body. This simplifies the research considerably. The results of the quarter car investigation revise the flat and spatial vehicle models.
- With the performance of the vehicle suspension elements is evaluated accurately, the test bench may be used for suspension improvement, for example, the development of a new active suspension.
- The vehicle suspension test bench can be used to investigate the performance of individual suspension elements by simulating various modes of vehicle movement and changing the load conditions.
- After assessing the lessons learned and installing a suspension of a different design, the test bench can be used to investigate other types of vehicle suspensions.

The test bench consists of the following main structural elements and assemblies.

- Main frame with road surface simulation equipment.
- Vehicle body part with suspension elements.
- Wheel drive with rotational speed changers.
- Equipment for recording the parameters describing the movement of the vehicle body.

The front part of the VW Golf-II vehicle with suspension elements was installed on the experimental test bench. During the investigation, the performance of the MacPherson suspension was analyzed. Various operating modes of the vehicle suspension were modeled by changing wheel rotational speed (four modes available); sprung mass of the body part, and the frequency of movement of the device simulating the effects of the road surface. The effect of the road surface was simulated by a pneumatic diaphragm chamber controlled by an electro-pneumatic distributor (Festo CPE 1 8-M1H-5/3G-1/4) that controlled the compressed air feed from the compressor. In the initial phase of the investigations, the road profile was modeled as a periodic function, and a discrete programmable controller (Festo FEC-FC30-DC) was used to control the distributor. For further investigation, microprocessor control of the distributor was provided according to real road surface microprofile records. The scheme of operation of the mechanisms that simulate the effects of the road surface and the quarter-car test bench is presented in Figure 14.

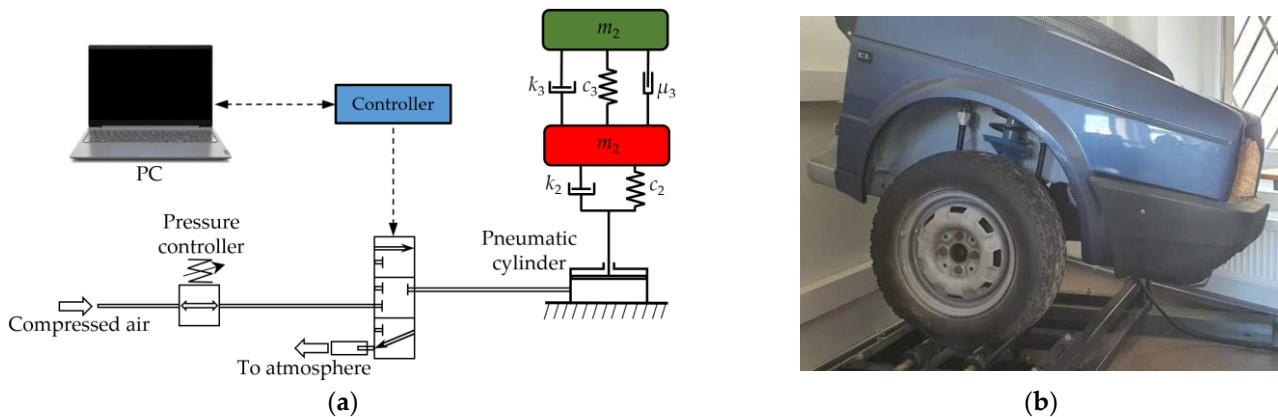

**Figure 14.** Equipment simulating the road surface effects: (**a**) scheme; (**b**) quarter-car test bench.

Oscillation analysis of individual assemblies of test bench assemblies was performed to investigate the movement of the vehicle body and suspension and to determine the main characteristics of the model. Oscillation measurement equipment was used to record the oscillation parameters (displacement, speed, and acceleration). To measure the oscillation signal, a two-channel oscillation recording converter ADC-212 'Pico' (Pico Technology, St. Neots, UK) was used together with software installed on the computer. For additional tests, and control of the results of previous tests, a portable oscillation recording device 'Multiviber' by VMI (VMI International AB, Linköping, Sweden) was used. The results of the additional investigations allowed us to compare the performance of a real vehicle suspension with the operating conditions simulated by the vehicle suspension test bench.

Movement of the vehicle body and suspension elements was investigated by measuring the vertical displacements and the nature of their change. Movements of the equipment that simulate excitation of the road surface were recorded by a sensor located on a moving frame arm. The effect of excitation on the movement of the unsprung mass of the suspension was recorded by a sensor located on the lower arm of the suspension of the wheel. Vehicle body movements were measured by a sensor located on the body next to the mounting joint of the MacPherson damper frame. This arrangement of the sensors allowed the researchers to register changes in the movements between the individual assemblies of the system determined by the elastic and damping properties of the suspension elements and the tire.

### 4.2. Analysis of Experimental Investigation Results

During the tests, the characteristics of free oscillations of the suspension of the vehicle were determined by applying a load on the body using an additional mass and abruptly removing the load (Figure 15). The relative damping coefficient of the suspension was determined on the basis of these characteristics in relation to the theoretical relative damping coefficient corresponding to the front suspension of the VW Golf vehicle. Comparison of the relative damping coefficient of the suspension determined on the test bench with the theoretically calculated one showed that the actually measured coefficient was slightly (15%) higher. This change may have been determined by frictional forces in the suspension elements that had not been considered.

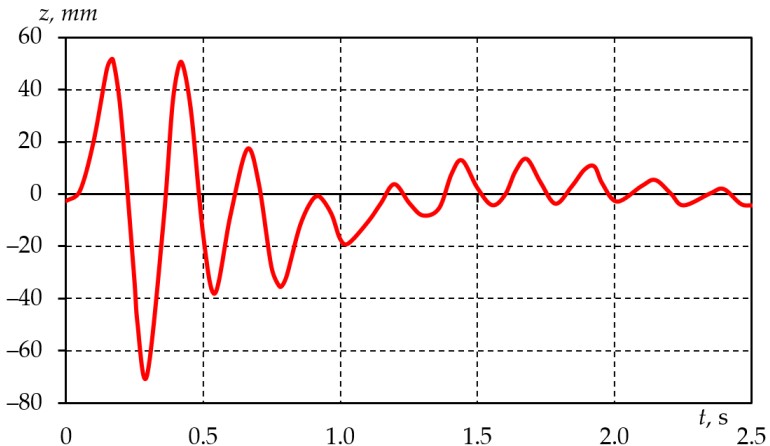

**Figure 15.** The characteristic of the free oscillations of the vehicle suspension was determined on the car suspension test bench.

Following measurement of the parameters describing the movement of the vehicle suspension elements, it was decided to use spectral records of oscillation displacement for the analysis of the results, as these records more accurately reflect the nature of low-frequency oscillations prevailing in the vehicle body and suspension elements. The spectral analysis of oscillations enabled a more accurate assessment of the components of oscillations of individual frequencies and the determination of their origin and the nature of the change. Investigation of vehicle body oscillations caused by wheel rotation (Figure 16) allowed the observation of the maximum values of the vertical displacements of the vehicle body corresponding to the wheel rotation speed.

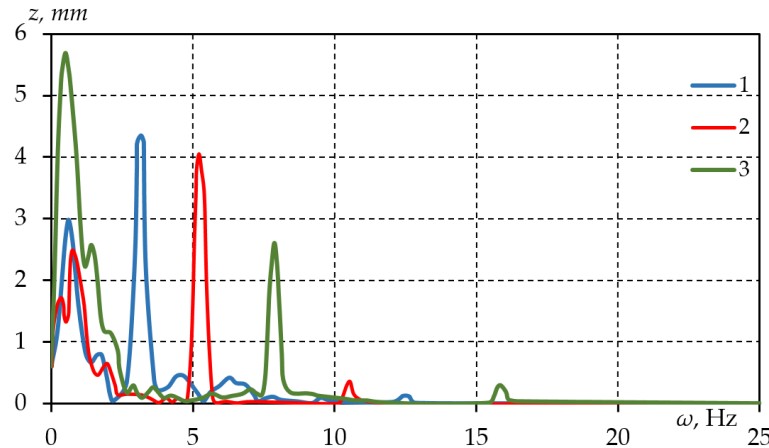

**Figure 16.** Spectrum of vehicle body displacements at different wheel rotational speeds: 1—gear I; 2—gear II; 3—gear III.

The frequency of these values corresponded to the rotational speed of the wheel set by the respective gear. The spectral record showed an evident increase in the vertical displacement of the 1 Hz frequency, which corresponded to the lower natural frequency of the resonance oscillations of the sprung mass. The amplitudes of the body resonance oscillations increased with increasing wheel rotational speed.

To verify the reality of the vertical displacements of the vehicle body measured on the test bench, real displacements of the vehicle body were measured when driving on a smooth asphalt pavement surface. Different low-frequency resonance displacements of the vehicle body were observed in the vertical displacement spectrum (Figure 17), and their amplitudes corresponded to the amplitudes measured on the test bench.

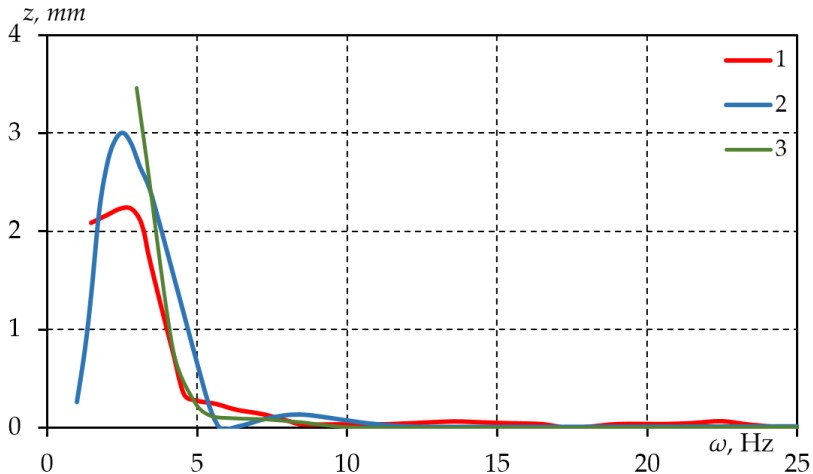

**Figure 17.** Spectrum of vertical displacements of the real vehicle body at different speeds: 1—23 km/h; 2—31 km/h, 3—56 km/h.

During the vehicle suspension investigation, the excitation of a rolling wheel of different frequencies was investigated, thereby simulating the effect of the road surface. Different peak displacement values were observed in the vertical displacement spectrum of the vehicle body (Figure 18).

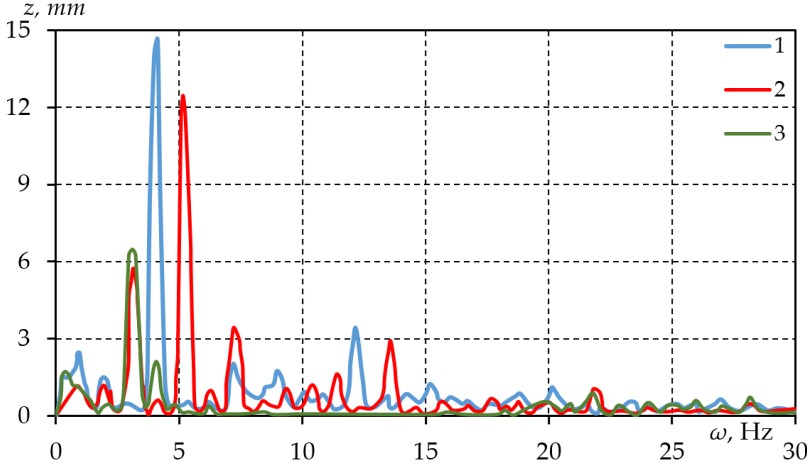

**Figure 18.** Spectrum of vertical displacements of the vehicle body determined by different excitation frequencies: 1—4.17 Hz; 2—5.55 Hz; 3—8.33 Hz; 4—25 Hz.

The frequency of the peak displacement values corresponded to the excitation frequency. The value of the vertical displacement determined by an excitation frequency of 25 Hz was not distinct. This may be due to the slow action of the pneumatic chamber that

simulates the effect of the road surface. Therefore, it can be concluded that it is reasonable to simulate only the low-frequency (<20 Hz) excitation effect of the road surface on the test bench. An experiment was performed to compare the vertical displacements of the suspension on the test bench and of the real vehicle body to verify the reliability of the test bench results. Spectral analysis of the vertical displacements (Figure 19) suggested that the values of the displacements and the nature of the change corresponded to each other. Existing inaccuracies may have been affected by an inaccurately simulated road surface, suspension elements, or the tire on the test bench. Fixing the car body to the test bench is another problem. A sufficiently rigid mounting can dampen the movement of the sprung mass of the model.

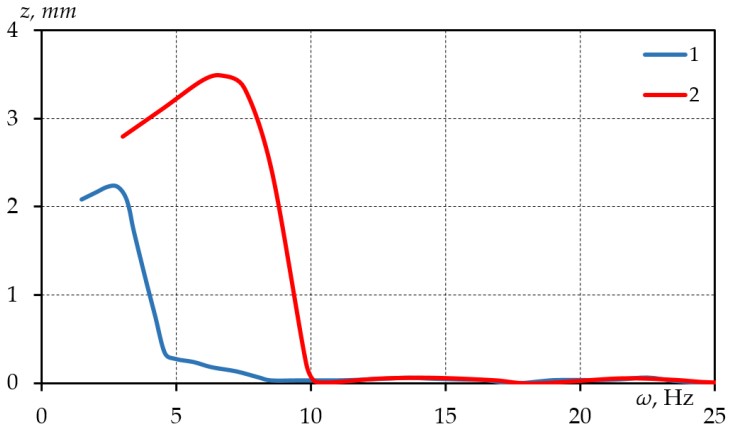

**Figure 19.** Comparison of vertical displacements of the vehicle body: 1—real vehicle; 2—suspension test bench.

## 5. The Effect of Vehicle Suspensions on Stability

### 5.1. Kinematic Model of Vehicle Suspensions

Three variants of vehicle suspension were modeled in the work: 1—front suspension: double arm Mercedes Benz type suspension (Figure 20a); 2—front suspension: double arm Honda type suspension (Figure 20a); 3—front suspension: MacPherson VW Golf type suspension (Figure 20b); 4—rear suspension: single-arm BMW type suspension (Figure 20c); 5—rear suspension: semi-dependent single-arm VW Golf type suspension (Figure 20c); 6—rear suspension—MacPherson Audi type suspension (Figure 20b).

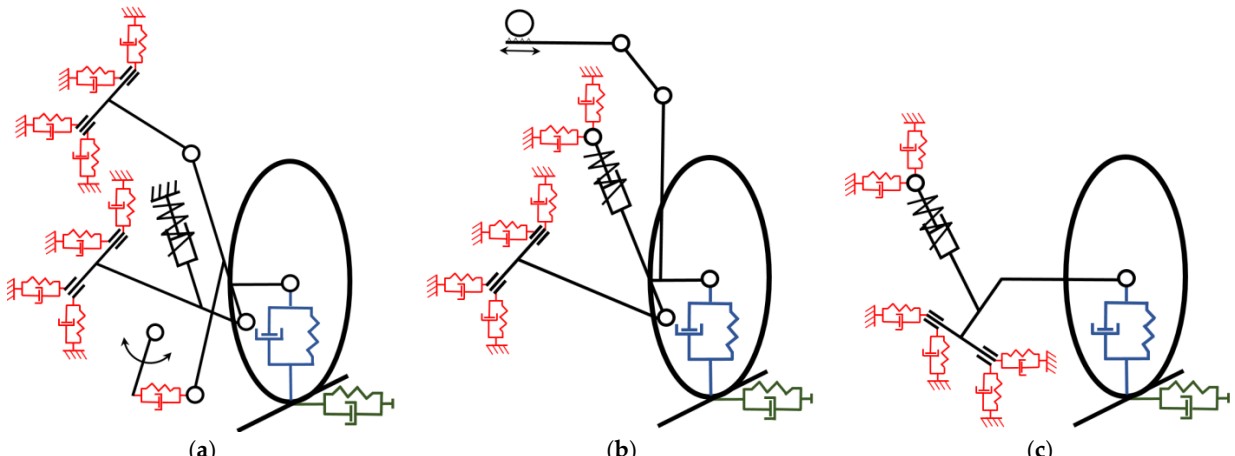

(a)          (b)          (c)

**Figure 20.** Kinematic schemes of the suspensions: (**a**)—double-arm suspension with lever steering mechanism; (**b**)—MacPherson suspension with gear-rack type steering mechanism; (**c**)—semidependent single-arm suspension.

Driving of the vehicle with each of the suspension combinations was simulated on three different road profiles (low quality pavement, even pavement, and gravel road) during the investigation. For this purpose, programs developed in Matlab were used. The main version was designed to study the kinematics of suspensions. The programs were developed on the basis of kinematic analysis of multiassembly mechanisms. In preparation for the use of the provided data for stability calculations, a calculation of the instantaneous centers of suspension on the transverse and longitudinal planes of the vehicle was introduced, and auxiliary values for the evaluation of the angular stiffness of the suspensions were determined. The parameters required for the calculation of the angular suspension stiffness values and the calculation results were transferred to the files and used directly for stability assessment programs. It was assumed that the vehicle was traveling at a constant speed of 90 km/h. The following characteristics of suspension performance were determined: wheel travel in the vertical direction $\Delta z$; wheel turning angle $\Delta \varphi$; wheel displacement in the transverse direction $\Delta y$ and wheel camber $\zeta$. The effect of the variation of these parameters on the stability was evaluated by using the generalized slip angle $\delta$:

$$\delta = \Delta \varphi + \delta_\zeta + \delta_y \tag{14}$$

where $\Delta \varphi$ is the change in wheel turning angle; $\delta_\zeta$ is the tire slip angle due to wheel camber and $\delta_y$ is the tire tread slip angle due to the transverse displacement of the wheel.

Tire slip angle $\delta_\zeta$ was determined as follows:

$$\delta_\zeta = \frac{\zeta}{k_\zeta} \tag{15}$$

where $\zeta$ is the wheel camber and $k_\zeta$ = 4–6.

Additional tire slip angles $\delta_y$ were calculated using the Equations:

$$\delta_y = \frac{F_y}{k_y} = \frac{\Delta y C_y}{k_y} \tag{16}$$

where $F_y$ is the load acting on the wheel in the transverse direction; $\Delta y$ is the transverse displacement of the wheel, $C_y$ is the transverse stiffness of the tire and $k_y$ is the slip angle drag coefficient.

Table 2 shows the vertical travel of the wheel $\Delta z$ with the car traveling on different road profiles. Based on these results, the suspension travel limits were set to calculate the changes in wheel position due to vertical suspension excitation.

**Table 2.** Vertical travel of the wheel (mm).

|  | 1 (Mercedes Benz Front) | 2 Honda (Front) | 3 VW (Front) | 4 BMW (Rear) | 5 VW (Rear) | 6 Audi (Rear) |
|---|---|---|---|---|---|---|
| Low Quality Pavement | 42.4 | 42.0 | 41.7 | 41.4 | 41.2 | 40.9 |
|  | −56.8 | −55.2 | −56.7 | −54.9 | −55.7 | −55.0 |
| Even Pavement | 14.1 | 14.0 | 13.8 | 13.0 | 13.0 | 12.8 |
|  | −14.1 | −13.9 | −25.7 | −13.8 | −13.8 | −13.4 |
| Gravel Road | 65.5 | 64.5 | 63.4 | 68.1 | 68.4 | 62.2 |
|  | −66.3 | −64.2 | −65.9 | −65.5 | −65.7 | −65.1 |

Note: "+" and "−" signs indicate the wheel travel up and down relative to the static position of the wheel, respectively.

Investigation of the kinematic characteristics of the suspensions showed that different road profiles had significant effects on suspension performance. When driving on low quality asphalt, the total suspension angles of the front suspension for the Mercedes Benz suspension ranged from −0.5 to 0.4°, for Honda—from −0.5 to 0.8°, and for VW for front suspension from −0.7 to 1.3°. For the rear suspensions, the slip angles were much smaller. For example, in the case of the BMW they varied from −0.005 to 0.03°, and for VW from

−0.01 to 0°. The front suspension of the Mercedes Benz and the rear suspension of the BMW were found to provide the best kinematic stability. The modeling results are presented in Figures 21 and 22.

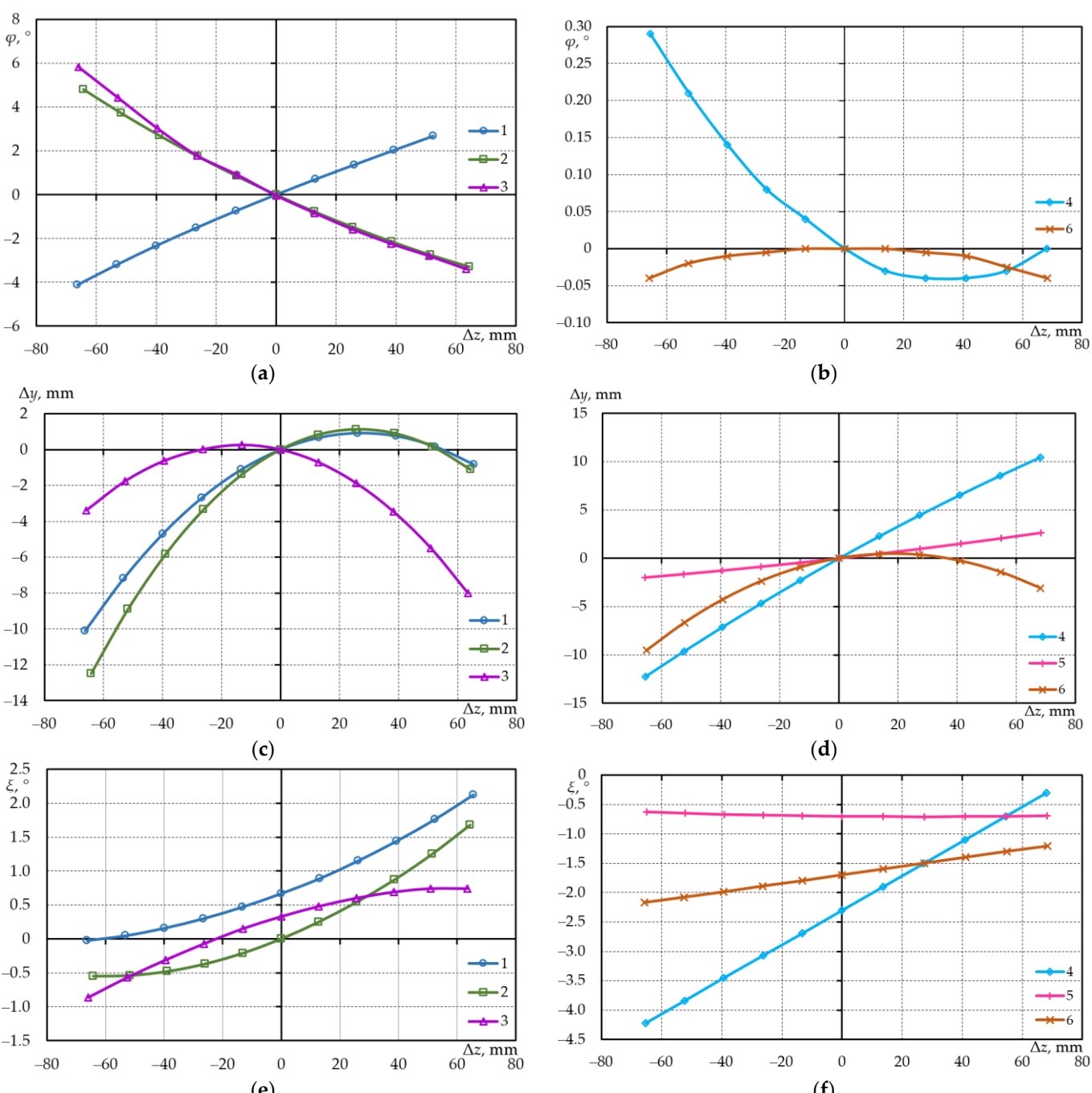

**Figure 21.** Dependence of front (**a,c,e**) and rear (**b,d,f**) suspensions on wheel position as a function of the vertical travel of the wheel: the turning angle of the wheels (**a,b**), transverse displacement of the wheels (**c,d**), wheel camber (**e,f**) (marking according to Table 2).

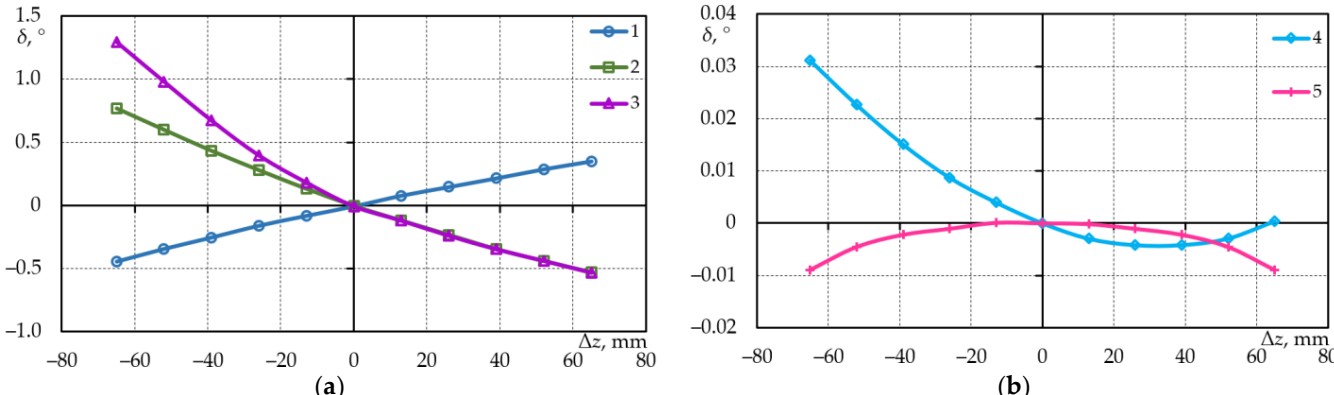

**Figure 22.** Dependence of the slip angle of the front (**a**) suspension and rear (**b**) suspension wheels on the vertical wheel travel (marking according to Table 2).

### 5.2. Investigation of the Dynamics of Change in Vehicle Direction

Following assessment of the kinematic properties of the suspensions, the suspension model was improved by introducing deformable elements at the characteristic locations of the suspension and steering mechanism. The deformability of the arms was evaluated in the suspension, and a nonrigid steering rod was used in the steering mechanism. All deformable elements of the suspension and steering mechanism were reduced to these elements (Figure 20a,b). To determine the spatial position of the wheels during variation of the vertical displacement, $z$ of the wheel, a coordinate system was used (Figure 23).

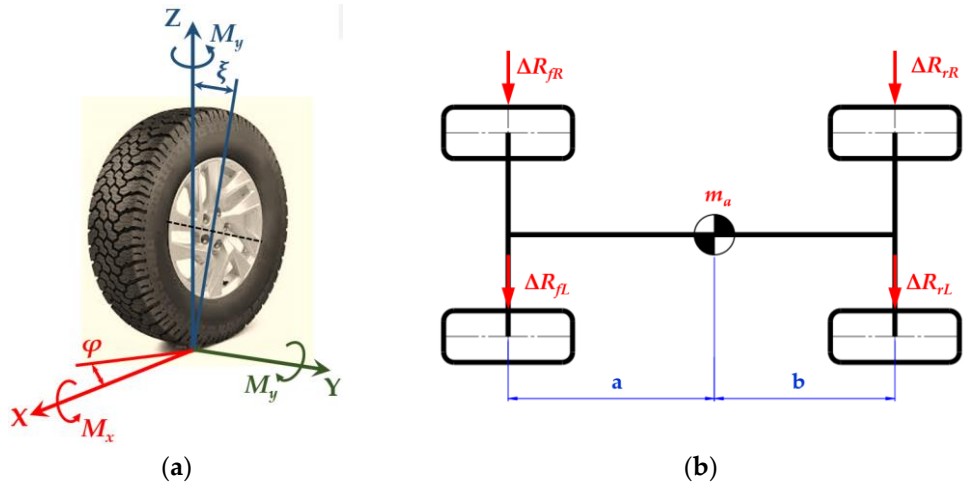

**Figure 23.** Wheel coordinate system of the 3D model (**a**); lateral forces acting on the wheels (**b**).

If the vehicle body is subjected to a lateral force, then the reaction resulting from transverse wheel displacements $\Delta y$ is:

$$R_{iy} = C_y \cdot \Delta y \qquad (17)$$

where $C_y$ is the transverse stiffness of the tire and $\Delta y$ is the transverse displacement of the wheel. Then:

$$m_a \cdot \ddot{y}_c = \Sigma \Delta R_{ih}, \quad I_{za} \cdot \ddot{\phi}_c = \left( \Delta R_{fL} + \Delta R_{fR} \right) \cdot a + \left( \Delta R_{rL} + \Delta R_{rR} \right) \cdot b \qquad (18)$$

When modeling vehicle stability, the effect of nonlinearity of the suspension on the vertical displacements of the body and wheels was investigated in view of the excitation frequency by combining different quality road profiles and vehicle speeds. The quarter-car

model was used (Figure 9). The driving of the vehicle on three different road profiles (low quality pavement, even asphalt pavement, and gravel road) was simulated during the investigation. The VW Golf vehicle was assumed to travel at a constant speed of 10, 20 and 30 m/s (36, 72, 108 km/h).

The following suspension performance characteristics were determined: wheel travel in the vertical direction $\Delta z$; wheel turning angle change $\Delta \varphi$; wheel displacement in the transverse direction $\Delta y$; wheel camber $\xi$ and wheel slip angle $\delta$ (Figures 24 and 25).

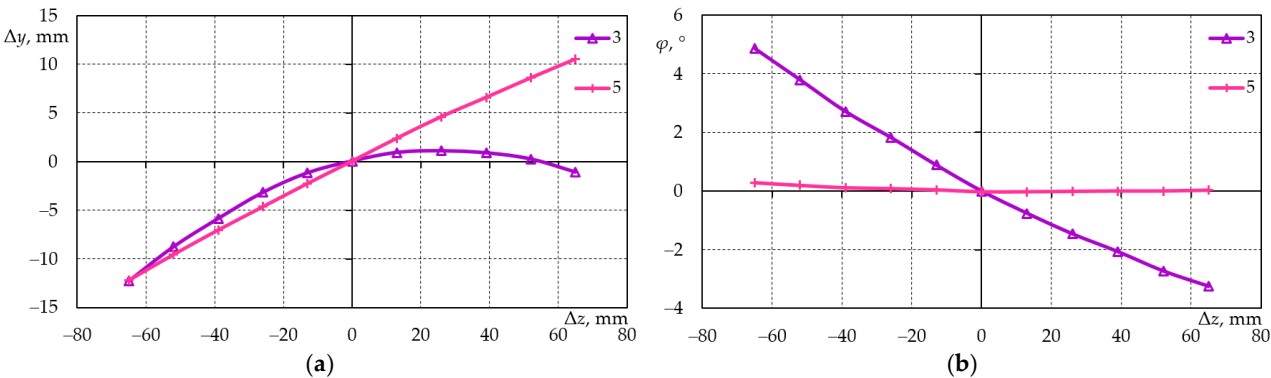

**Figure 24.** Dependence of transverse wheel displacement (**a**) and wheel turning angle (**b**) on vertical wheel travel (marking according to Table 2).

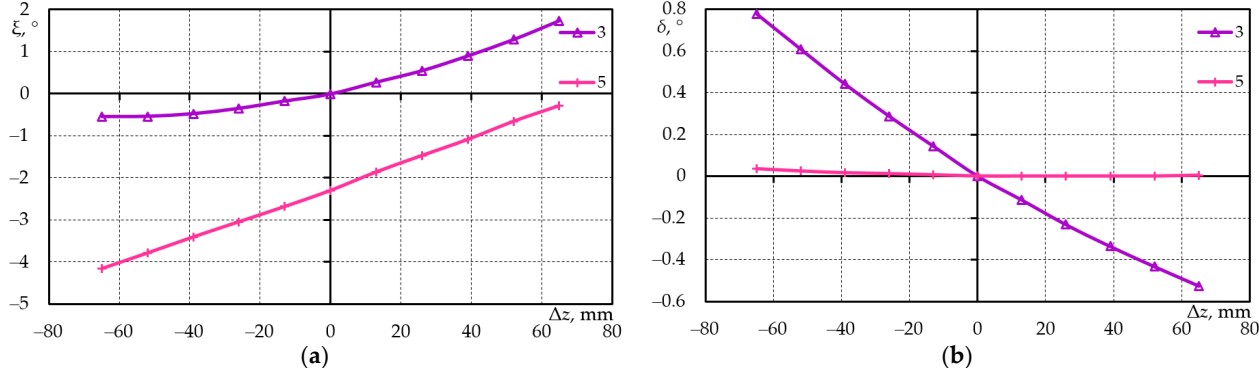

**Figure 25.** Dependence of wheel camber (**a**) and slip angle (**b**) on vertical wheel travel (marking according to Table 2).

The Matlab program for the investigation of suspension kinematics was used for description of values $\Delta \varphi$, $\Delta y$, $\xi$. The results obtained that describe the dependence of the parameters of the mentioned parameters on the vertical travel of the wheel are presented in Figure 26.

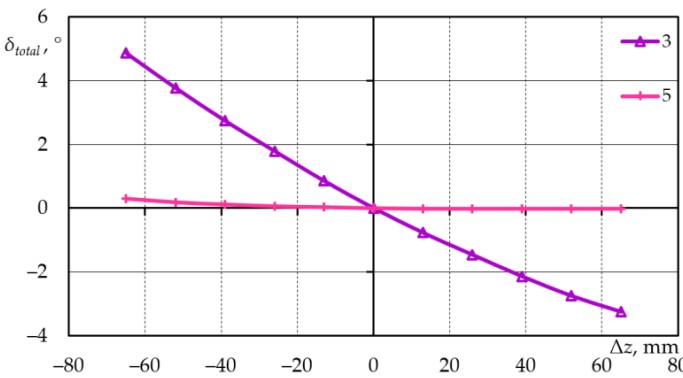

**Figure 26.** Dependence of total slip angle on vertical wheel travel (marking according to Table 2).

The results obtained show that additional investigations of vehicle stability are needed in the dynamic mode, i.e., when the car is moving, as the total slip angles obtained exceed 4 degrees in borderline cases. Previous vehicle stability calculations show that these angles can be critical under transverse stability conditions even at relatively low speeds (up to 100 km/h). To this end, additional investigations on the vertical dynamics of the vehicle were performed to evaluate the influence of nonlinear suspension elements on the tire tread and the dynamics coefficients of unsprung mass and sprung mass:

$$k_d = \frac{z_i}{q} \tag{19}$$

where $z_i$ is the $i$-th mass displacement; $q$ is the height of the unevennesses ($q$ is assumed to be equal to the average height of the unevenness corresponding to the even asphalt pavement road surface).

*5.3. Cornering Stability of the Vehicle*

Modeling of the linear motion according to the equations provided in Section 5.1, including additional evaluation of the changes in the spatial position of the wheel, showed the necessity to input the driver's reaction. Without the driver, such modeling is not effective because the car changes direction due to accidental effects and requires frequent correction to simulate direction adjustment using the steering mechanism. However, in this work, the technical specifications for the model with a driver were prepared, while the excitation parameters when driving on roads of different quality could be established using this model. The developed Matlab programs enabled the determination of suspension deformations (and tire-road contact geometry for more accurate modeling).

The appropriateness of the solutions was evaluated using a more defined model that examined vehicle behavior under stabilized conditions, i.e., when moving in circles.

Vehicle cornering stability was investigated but the studies did not take into account the change in the position of the wheels during vehicle tilt, which may affect vehicle stability. In this work, the vehicle behavior model was refined by evaluating the kinematics of the vehicle's front and rear axles and their influence on the vehicle's behavior.

When examining vehicle cornering stability, two criteria are usually evaluated [50]: loss of stability when the car slips and loss of stability when the car rolls over. Our study additionally specified the extent of the change in the vehicle at cornering due to suspension deformations and changes in the position of the wheels at car rollover.

The main problem encountered in the evaluation of vehicle stability at corners was that the lateral inertia forces redistributed the vertical wheel loads. The solution to the problem was initiated by investigating the problem of car tilt in cornering. In the programs for the kinematic analysis of the suspension, the position of the center of instantaneous tilting was additionally determined for the front and rear axles. The application with the subroutine provides additional data required to determine the tilt angle, the instantaneous tilt centers, the angular stiffness coefficients of the suspension, and stabilizers.

In determining the distribution of vertical reactions, the assumption was made that under the action of a lateral force, the sprung part of the vehicle would rotate about the axis connecting the tilt centers of the two axles. When calculating vehicle tilt, simplified tilt schemes (Figure 27) are usually chosen, which do not take into account body (frame) deformations and changes in the center of position of the tilt center during vehicle tilt.

With acceleration in the lateral direction equal to $\mu$, the damping mass $m_a$ of the vehicle would be subject to force $m_a\mu$. If the designations in Figure 27 are used, the distance from the road surface to the tilt axis at the center of mass is equal to:

$$h_p = h_{1L} + (h_{2L} - h_{1L})/L = (h_{2L}a_1 + h_{1L}a_2)/L \tag{20}$$

The body tilt is equal to:

$$\beta = \frac{\Delta R_1 + \Delta R_2}{\frac{(c_{1k}+c_{1sk})c_{kpa}}{c_{1k}+c_{1sk}+c_{kpa}} + \frac{(c_{2k}+c_{2sk})c_{kpa}}{c_{2k}+c_{2sk}+c_{kpa}}} t \tag{21}$$

where $\Delta R_1$, $\Delta R_2$ are the changes in radial loads on the front and rear axles, respectively, due to the lateral force, and $c_{ik}$, $c_{isk}$, $c_{kpa}$ are the coefficients of angular stiffness of the suspension, angular stabilizer and the tires, respectively.

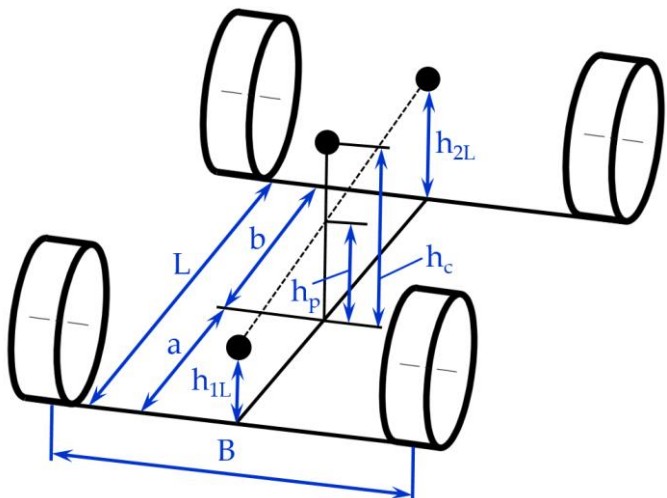

**Figure 27.** Wheel load recalculation scheme when the vehicle is subjected to lateral force.

The tilt of the front and rear wheels is equal to:

$$\beta_1 = \beta \frac{c_{kpa}}{c_{1k} + c_{1sk} + c_{kpa}}, \; \beta_2 = \beta \frac{c_{kpa}}{c_{2k} + c_{2sk} + c_{kpa}} \tag{22}$$

The angle of slipping of the wheel may be calculated after determining the tilt of the wheels. As already mentioned, changes in the camber angle of the wheel and the turning angle of the wheel about the vertical axis were evaluated by adjusting the kinematically obtained turning angle of the wheel, which was assumed to be the change in the angle of slip.

Calculations were performed by estimating the limit turning radius $R_{rib}$ for different speeds. This was calculated according to the grip conditions (turning radius limited by the car slip or camber) with nondeformable and deformable suspension (grip coefficient $\mu = 0.8$). The results are presented in Table 3. The additional body rollover angle can be observed to undergo no significant change due to suspension deformations under these conditions.

**Table 3.** Vehicle boundary turning radius.

| Speed, km/val | Turning Radius at Slip, $R_{ribS}$, m | Turning Radius at Rollover, $R_{ribV}$, m | |
|---|---|---|---|
| | | Deformable Suspension | Non-Deformable Suspension |
| 20 | 3.50 | 1.80 | 1.80 |
| 40 | 13.98 | 7.20 | 7.21 |
| 60 | 31.46 | 16.20 | 16.23 |
| 80 | 55.93 | 28.81 | 28.86 |
| 100 | 87.39 | 45.01 | 45.09 |
| 120 | 125.85 | 64.82 | 64.92 |
| 140 | 171.29 | 88.23 | 88.37 |
| 160 | 223.73 | 115.23 | 115.42 |

In further investigations, a change of direction was noted to cause a transverse inertia force, which led to the redistribution of radial reactions. The kinematics of the suspension, in turn, led to possible changes in the spatial position of the wheels.

Since the slip angle depends on the radial loads, in particular under boundary conditions, the redistribution of the radial loads must be evaluated when solving stability problems. For this, the algorithm provided the determination of the reaction $R_y$ of each tire according to it load.

The AUTOSTAB program was developed for the implementation of the model. The algorithm is shown in Figure 28.

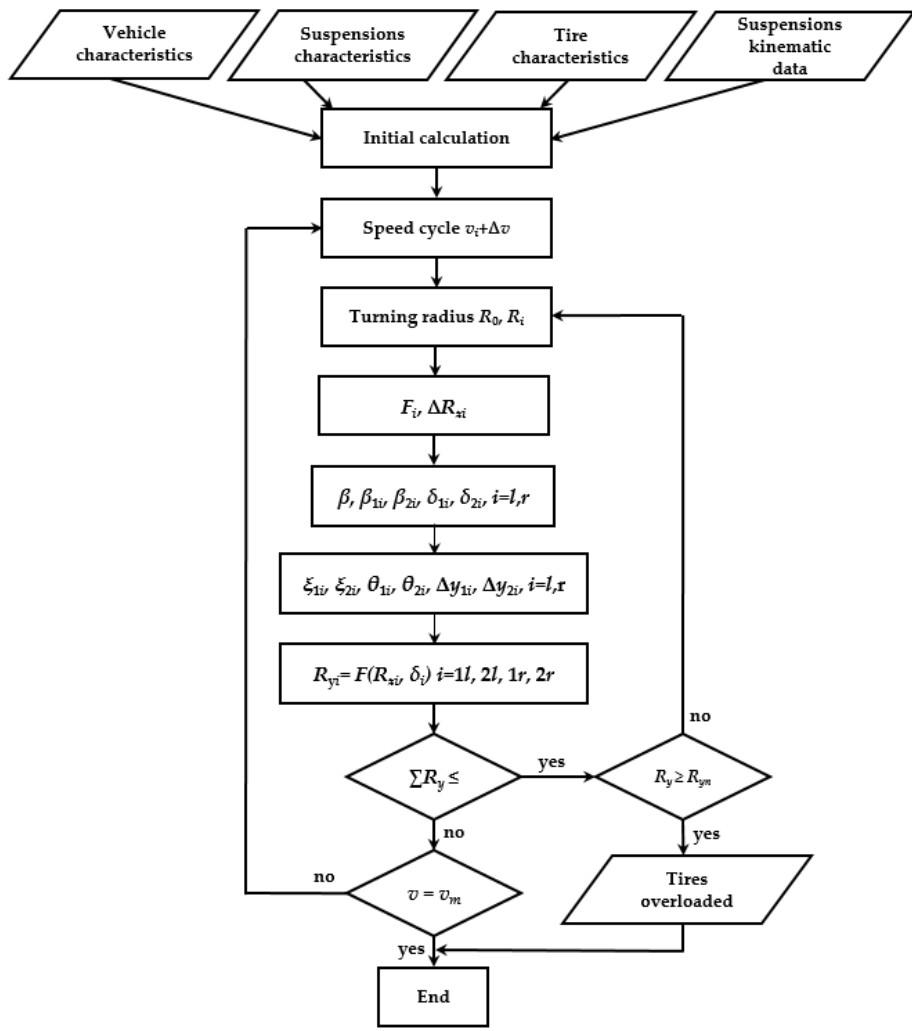

**Figure 28.** Vehicle boundary turning radius calculation program AUTOSTAB algorithm.

The simulation results show that in modeling the directional stability of the vehicle, it is necessary to evaluate the specifics of the suspension, the redistribution of radial loads under the action of lateral force and changes in the geometric position of the wheel due to the kinematic properties of the suspension.

The results obtained by modeling the movement of the car under stabilized conditions are presented in Figure 29.

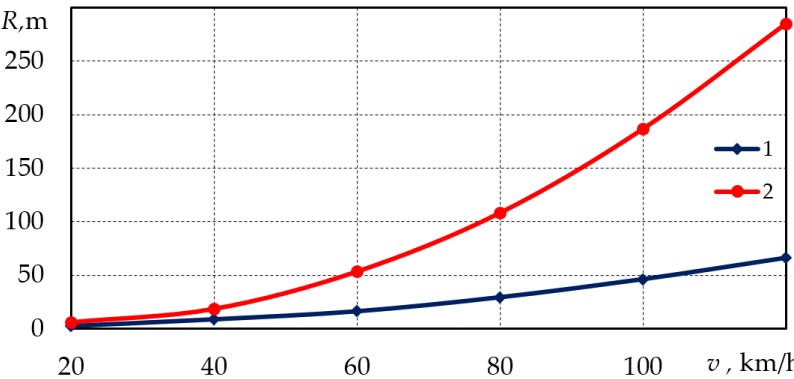

**Figure 29.** Dependence of the boundary turning radius on the vehicle speed with (1) and without (2) the distribution of radial loads, and the change in wheel position during deformation of the suspension.

Figure 29 shows that by estimating the distribution of radial loads and changes in wheel position, the vehicle boundary turning radius increased up to 4.39 times at 120 km/h speed.

The developed methodology enabled an evaluation of the influence of road condition on directional stability under vertical excitation, investigation of the sensitivity of individual suspension types to vertical excitation according to directional stability criteria, and an assessment of the technical condition of the vehicle if patterns of change in suspension parameters are available. The technical condition was assessed in the work by changing the stiffness of the suspension elements.

Application of the numerical model to linear movement showed that this kind of modeling would not be effective without a driver's model. This was due to the fact that the vehicle would change direction under the action of random effects, and frequent correction would be required to simulate direction adjustment using the steering mechanism. Technical specifications for this kind of model were prepared in this work, and the excitation parameters when driving on roads of different quality may be established using this model.

## 6. Discussion

The investigation showed that when modeling the effect of the road surface using a flat model, the effect of the optimum evaluation of the road surface effect could be provided using microprofile records on one longitudinal line. The spatial records of the road surface parameters reflect the properties of the road surface in the transverse direction. However, for a thorough evaluation of the interaction, small-interval microprofile data must be recovered from more accurate records. In the above cases, the roughness of the road surface could be assessed. Investigations have suggested that the IRI measurement methodology proposed by M. W. Sayers for measuring the microprofile every 0.147 m provides an adequate reflection of the road-vehicle interaction characteristics, as it corresponds to the average length of the passenger car tire-road contact, which allows accurate prediction of road surface effects. On the basis of the results of the tire comparison function investigations, it can be concluded that the description of a tire with an elementary filter that compares waves of a certain length is not efficient. Rough surface unevennesses with amplitudes greater than 4 mm were found to be not always absorbed by tires. The tire can absorb a single random unevenness, or a repetitive set of them, otherwise the roughness is registered as micro-irregularity and causes additional wheel loads. Therefore, in more accurate models, not only the road microprofile, but also the roughness of the road surface needs to be assessed. After examining various methods of evaluation of the tire comparison function, it was determined that when calculating the uniformity of vehicle travel, the data from spatial road microprofile records can be recalculated by comparing properties of the tire. These data are then used to investigate suspension performance, in which case the

tire-road interaction is described by the point contact model. It was determined that where data on wheel tilt or contact curve on the plane transverse relative to the wheel roll are unnecessary, the use of the 3D tire model would also be unreasonable. The vertical wheel displacement required for the evaluation of suspension performance may be determined by the tire model with point contact. Programs were developed that analyze the movement of the quarter-car model on road surfaces of various characteristics. In order to increase the accuracy of calculations, it was necessary to evaluate the nonlinearity of vehicle suspension elements, friction in suspension elements, and the effect of tire tread properties on suspension performance, in particular when modeling driving on low quality roads or when assessing gross, random road surface damage.

To verify the theoretical calculations characterizing vehicle suspension, and to investigate various suspension parameters, a vehicle suspension test bench corresponding to the quarter-car model was designed and manufactured. A mechanism of the original design was developed to simulate the effect of the road surface, which allowed the modeling of various rolling conditions of the wheels. After evaluating the results of the investigation carried out on the experimental test bench, comparing theoretical and practical data, it can be concluded that the results obtained corresponded with reality. This was confirmed by analysis of the free oscillation characteristics of the vehicle suspension measured on the test bench and the correspondence of the relative damping coefficient of the suspension with that determined by theoretical calculations (the results obtained differed by 15%). Spectral analysis of the vertical displacement of the vehicle body and other suspension mechanisms measured on the suspension test bench confirmed correspondence of the properties of the quarter-car model with corresponding characteristics of the real vehicle. Therefore, it can be stated that the real quarter-car model corresponded to the theoretical model, as well as to real car movement. The developed model is appropriate for further investigation and modeling of the suspension properties of cars. Minor inaccuracies were also observed, which may have affected the reliability of the results. The limited capacities and slowness of the pneumatic chamber simulating the excitation of the road surface, which allowed testing only at a low-frequency (<20 Hz) of excitation should be mentioned. Another issue was the attachment of the vehicle body to the test bench frame, as a sufficiently rigid attachment may inhibit the movement of the model sprung mass. However, the conditions simulated by the test bench offered a fairly accurate reflection of the real conditions; therefore, it is reasonable to further investigate the characteristics of the real quarter-car module, as well as possibilities for its improvement.

A vehicle stability model was developed for the assessment of vehicle-road interaction. The model comprehensively evaluated the dynamic characteristics of the suspension, suspension kinematics, and road conditions, and enabled the development of methodology for the determination of the safe speed of the vehicle on roads of known quality using numerical models.

As a result of the combination of the vertical displacement of the quarter car and the kinematic models, a suspension model was developed and described changes in the spatial position of the wheel during suspension operation. When driving on roads of different quality, the data obtained allowed estimation of the directional stability of the vehicle, including the lateral displacement of the wheel, camber, and the turn about the vertical axis. To simplify the analysis, all these changes were reduced to one parameter, i.e., the calculated tire slip angle.

Simulations showed that the kinematic properties of the front suspension needed to be refined in the directional simulation application, as the boundary travel of the suspension resulted in large total slip angles (in particular, the front wheels). This causes the risk that the vehicle may lose its transverse stability at higher speeds.

The effect of changes in the geometric position of the wheel on the directional stability parameters of the vehicle was determined using kinematic models. The total slip angle was used for this. The software application AUTOSTAB was developed under this methodology

taking into account the effect of the above factors on the dependence of the vehicle's limit turning radius on speed.

Following the example of the VW Golf, changes in the geometric position of the wheel during suspension operation were found to have a significantly greater effect on directional stability than body tilt due to centrifugal forces.

## 7. Conclusions

1.  A comprehensive analysis of vehicle-road interactions shoed that in order to determine the stability of vehicle movement it is necessary to accurately assess the characteristics of the interacting subsystems. Factors influencing vehicle-road interactions were specified during the investigation, including road surface characteristics; nature of road surface, tire contact area and compared properties of the tire, suspension design; kinematic and dynamic models of the suspension, all of which affect stability of the vehicle movement.

2.  A revised model of tire equalizing function was developed. The coordinates of relative measures were found to cause no essential differences in the assessment of tire deformations of different vehicle tires. This means that the number of tire groups may be reduced to evaluate their impact on stability.

3.  A dynamic quarter-car model was developed for a comprehensive integrated assessment of suspension characteristics, suspension kinematics, and road surface condition. The model enabled the development of a methodology for the determination of the safe speed of the car on roads of known quality using numerical models.

4.  Analysis of the revised quarter-car model showed that the nonlinearity of the suspension elements could be ignored only with examination of movement on a good quality asphalt pavement, because the suspension elements work in the linear part of the suspension characteristic. The effects of nonlinear elements occurred when driving on low quality roads and in case of severe, random damage to the road surface. During large suspension movements, it was necessary to account for friction forces in the damper frame; therefore, methodology for formalizing friction force was considered.

5.  The adequacy of the theoretical characteristics of the quarter-car model in relation to the real characteristics was investigated using an experimental test bench. The relative damping coefficient of the vehicle suspension measured on the test bench was approximately 15% higher than the calculated coefficient. The frequency characteristics of the vertical displacements of the real vehicle and vehicle body measured on the test bench were similar to those predicted by the quarter-car model in terms of the magnitude and nature of the change. The developed model is appropriate for further investigations and improvement of vehicle suspension properties.

6.  The revised kinematic model of suspension confirmed that the relative real camber changes by up to 2° when modeling an asphalt road surface compared to an ideally horizontal road surface. This causes the alignment angle of the wheel to change by up to 1.5°.

7.  The results of the experiment on the asphalt pavement road surface were compared with the model. Certain issues were observed in the analysis of experimental results due to additional oscillations and vibrations occurring in the body and suspension elements, which could be avoided by modeling movement after recording the road microprofile with a profilograph. The parameters that had the greatest impact on the stability of the direction—the difference between the displacement of the sprung and unsprung masses in the model versus the field experiment—did not exceed 15%.

8.  Changes in car behavior at changing speeds are related not only to the slip effect but also to changes in wheel spatial position due to suspension kinematics and suspension incompatibility with the steering mechanism. This effect can be particularly pronounced when road defects (pits, ruts) lead to an increase in suspension movement.

**Author Contributions:** Conceptualization, V.L., R.M., A.R., R.K., A.D. and R.S.; methodology, V.L., R.M., A.R., R.K., A.D. and R.S.; software, V.L., R.M., A.R., R.K., A.D. and R.S.; validation, V.L., R.M., A.R., R.K., A.D. and R.S.; formal analysis, V.L., R.M., A.R., R.K., A.D. and R.S.; investigation, V.L., R.M., A.R., R.K., A.D. and R.S.; resources, V.L., R.M., A.R., R.K., A.D. and R.S.; data curation, V.L., R.M., A.R., R.K., A.D. and R.S.; writing—original draft preparation, V.L., R.M., A.R., R.K., A.D. and R.S.; writing—review and editing, V.L., R.M., A.R., R.K., A.D. and R.S.; visualization, V.L., R.M., A.R., R.K., A.D. and R.S.; supervision, V.L., R.M., A.R., R.K., A.D. and R.S.; project administration, V.L., R.M., A.R., R.K., A.D. and R.S.; funding acquisition, V.L., R.M., A.R., R.K., A.D. and R.S. All authors have read and agreed to the published version of the manuscript.

**Funding:** This research received no external funding.

**Institutional Review Board Statement:** Not applicable.

**Informed Consent Statement:** Not applicable.

**Data Availability Statement:** Not applicable.

**Conflicts of Interest:** The authors declare no conflict of interest.

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
