# Peer review of "Investigation of Vehicle Stability with Consideration of Suspension Performance"

_applsci, doi:10.3390/app11209778_

Round 1

Reviewer 1 Report

The authors show different studies on the analysis of road surfaces, suspension kinematics and vehicle cornering stability in their manuscript. First an introduction is given with a literature review. Unfortunately standard literature on the topic, e.g. Tyre and vehicle dynamics (Pacejka), is not cited. After that, some basic information on vehicle modeling is given.

In the following section assessment of road surface properties is discussed as well as the properties of the tire road contact patch. Then, a quarter car model is introduced, which rather belongs to section 2.

In the fourth section an experimental test bench is shown, which should be used to analyze real road surface roughness on the vehicle suspension under laboratory conditions. Compared with real vehicle measurements (Figure 19) it can be seen that there is a large discrepancy compared to the test bench results. The authors say "The existing inaccuracies may have been affected by the inaccurately simulated 596 road surface effects on the test bench.", but other factors, e.g. different conditions of the suspension elements or tires, are not mentioned.

The next section covers the influence of suspension kinematics and elasticity on wheel position using three different examples of commercial vehicle's suspensions. No detials on modeling or modelparameters are given. Based on this, the curve radii, where the vehicle starts slipping or rollover occurs, for different velocities are detmerined by simulation. An influence of the suspension kinematics and elasticity could be found. Also the influence of road condition and technical condition of the suspension are mentioned at the end of the section, but results are missing.

The manuscripts ends with a detailed discussion and conclusion.

In summary the main goal of the investigation is to show, that taking the suspension kinematics into account is essential when analyzing vehicle stability. The novelty of this conclusion does not seem to be given. Further, the chapters on the assessment of the road surface properties and on the quarter car suspension test bench seem poorly integrated in this context. Also, capable vehicle suspension test benches and multibody vehicle dynamic simulation software, including complex suspension kinematics, are state of the art and commercially available. At last, the readability of the manuscript is partly made difficult by the writing style (e.g. Line 558 "Investigation of the oscillations of the vehicle body caused by the wheel rotation (Figure 16) enabled observation of the peak values of the vertical displacements of the vehicle body corresponding to the wheel rotational speed."). Often established technical terms are not used or replaced by others (e.g. Figure 8 "The frequency of tire contact length distribution [...]", Line 720 "[...] the loss of stability when at the car slip [...]").

Reviewer 2 Report

This study attempts to investigate the vehicle stability considering the performance of vehicle’s suspension system. Specifically, the effect of road-tire interaction and suspension system on the movement stability are studied. The features of suspension system are also validated by experiments. Overall, the paper is well written and the methodology is sound. The reviewer has several comments.

  1. The literature review should be more concise and organised, as the current version is not very easy to follow and the gap of the state-of-art is not clear.
  2. Regarding the experimental results, as they are not consistent with the theoretical results in terms of both the relative damping coefficient and vertical displacement, which result should be considered as the ground truth and how the inconsistency affects the following analysis.
  3. The application of the developed models is not clearly presented, i.e., how could they be used for improving the stability of vehicle.
  4. The paper needs careful proofreading, while there exist some grammar errors (such as ‘is focused on’, ’additionally investigation’, etc.) and the units of some tables are missing.

Round 2

Reviewer 1 Report

Thank you for revising the manuscript. I propose to accept the manuscript for publication.